# Synaptic activity regulates AMPA receptor trafficking through different recycling pathways

Ning Zheng[1,2], Okunola Jeyifous[1,3], Charlotte Munro[4], Johanna M Montgomery[3,4], William N Green[1,3]*

[1]Department of Neurobiology, University of Chicago, Chicago, United States; [2]Department of Molecular Genetics and Cell Biology, University of Chicago, Chicago, United States; [3]Marine Biological Laboratory, Woods Hole, United States; [4]Department of Physiology, University of Auckland, Auckland, New Zealand

**Abstract** Changes in glutamatergic synaptic strength in brain are dependent on AMPA-type glutamate receptor (AMPAR) recycling, which is assumed to occur through a single local pathway. In this study, we present evidence that AMPAR recycling occurs through different pathways regulated by synaptic activity. Without synaptic stimulation, most AMPARs recycled in dynamin-independent endosomes containing the GTPase, Arf6. Few AMPARs recycled in dynamin-dependent endosomes labeled by transferrin receptors (TfRs). AMPAR recycling was blocked by alterations in the GTPase, TC10, which co-localized with Arf6 endosomes. TC10 mutants that reduced AMPAR recycling had no effect on increased AMPAR levels with long-term potentiation (LTP) and little effect on decreased AMPAR levels with long-term depression. However, internalized AMPAR levels in TfR-containing recycling endosomes increased after LTP, indicating increased AMPAR recycling through the dynamin-dependent pathway with synaptic plasticity. LTP-induced AMPAR endocytosis is inconsistent with local recycling as a source of increased surface receptors, suggesting AMPARs are trafficked from other sites.

*For correspondence: wgreen@bsd.uchicago.edu

**Competing interests:** The authors declare that no competing interests exist.

## Introduction

NMDA- and AMPA-type glutamate receptors (NMDARs/AMPARs) are the major excitatory synaptic receptors in brain. They are held at post-synaptic densities (PSDs) by scaffold proteins aligning the receptors with the presynaptic glutamate release sites. Changes in synaptic strength, such as long-term potentiation (LTP), long-term depression (LTD) (*Malinow and Malenka, 2002*), and homeostatic plasticity (*Pérez-Otaño and Ehlers, 2005*), largely reflect the number of functional synaptic AMPARs. AMPAR internalization and recycling regulates AMPAR levels at synapses. Other processes, including diffusion of extrasynaptic AMPARs outside PSDs, association and dissociation of AMPARs with PSDs, and the number of 'slots' that AMPAR can occupy in PSDs (*Malinow and Malenka, 2002*; *Ehlers, 2000*; *Lin et al., 2000*; *Petrini et al., 2009*; *Ehlers et al., 2007*), also contribute to setting AMPAR levels at PSDs (*Opazo et al., 2012*).

During synaptic stimulation, 'constitutive' AMPAR recycling is increased several fold to become 'activity-dependent' AMPAR recycling (*Ehlers, 2000*). AMPARs undergo endocytosis through clathrin-coated pits during activity-dependent recycling (*Carroll et al., 1999*; *Lüscher et al., 1999*) and before exocytosis, traffic through the recycling endosomes (REs) identified by co-localization with transferrin receptors (TfRs) (*Ehlers, 2000*) and Rab11 (*Park et al., 2004*). During LTP, REs move from the dendritic shaft into synaptic spines (*Park et al., 2006*) from which regulated exocytosis of AMPARs appears to occur. It is uncertain whether AMPARs are exocytosed outside of the spines and traffic to

**eLife digest** Cells called neurons transmit information around the brain in the form of electrical signals. At a junction between two neurons—called a synapse—an electrical signal triggers the release of small molecules called neurotransmitters. These molecules travel across the gap between the two neurons and trigger a new electrical signal in the second neuron. Memories can be stored in synapses: high levels of activity can 'strengthen' the synapse, which increases the transfer of information between the neurons.

In many synapses, a molecule called glutamate is the neurotransmitter. Proteins called AMPARs, which are found on the surface of the neuron, can detect glutamate and transmit the signal along the second neuron. The strength of synapses is controlled by changes in AMPAR levels through 'recycling', where AMPAR proteins are removed from synapses, internalized and later returned to synapses.

It was thought that AMPARs are recycled via just one pathway at synapses. However, the amount of recycling is much higher when the synapses are active and it is not clear how this works. Now, Zheng et al. have used fluorescent tags to track the recycling of AMPARs in synapses from rats under a microscope. The experiments show that when the synapses are not active, most AMPARs are recycled via a pathway marked by a protein called Arf6. However, when the synapses are active, most AMPAR is recycled via a different route marked by so-called 'transferrin receptor' proteins.

The experiments also reveal that a protein called TC10 is involved in recycling AMPARs alongside Arf6, but is not required for recycling when the synapses are active and being strengthened. Unexpectedly, AMPAR internalization—via the process involving transferrin receptors—increases during synapse strengthening. This suggests that some of the extra AMPAR proteins sent to the membrane have come from other parts of the neuron away from the synapse.

Zheng et al.'s findings provide evidence that AMPARs are recycled through different routes depending on the activity of the synapse. The next challenge will be to directly test whether AMPARs are transported from other parts of the neuron to the strengthened synapse and to understand how this works.

PSDs via lateral diffusion (*Ashby et al., 2006*; *Yudowski et al., 2007*; *Tao-Cheng et al., 2011*; *Malinow and Malenka, 2002*) or are exocytosed at specific sites near PSDs (*Mohanasundaram and Shanmugam, 2010*). A minimal model of AMPAR constitutive and activity-dependent recycling has emerged from these and other studies. First, a single recycling pathway is assumed that starts at clathrin-coated pits (*Blanpied et al., 2002*), moves through REs, and ends with exocytosis back at the plasma membrane. Second, it is assumed that AMPAR recycling occurs locally, that is, AMPAR endocytosis and exocytosis occur at sites within the same synaptic domain. The model predicts that during LTD-AMPAR levels in recycling, endosomes and/or lysosomes at PSDs increase because endocytosis increases without increasing exocytosis from REs (*Ehlers, 2000*; *Fernández-Monreal et al., 2012*). During LTP, endocytosis is unchanged and AMPAR levels at PSDs are predicted to increase because of their exocytosis from REs causing decreased levels of AMPARs in REs (*Park et al., 2006*; *Groc and Choquet, 2006*).

Many factors specifically affect AMPAR activity-dependent recycling without affecting constitutive recycling [e.g., AP2 (*Lee et al., 2002*), Brag2 (*Scholz et al., 2010*)], and vice versa [e.g., NSF (*Lee et al., 2002*), PIP3 (*Arendt et al., 2010*)]. These studies suggest that AMPAR activity-dependent recycling is uncoupled from constitutive recycling and that separate processes underlie the two types of AMPAR recycling. Transmembrane AMPA receptor regulatory proteins (TARPs), such as stargazin, interact with recycling AMPARs at synapses (*Tomita et al., 2004*; *Neudauer et al., 1998*). TARPs interact with neuronal isoform of PDZ-protein interacting specifically with TC10 (nPIST) (*Cuadra et al., 2004*), suggesting that the Rho small G protein, TC10, might be a regulator of AMPAR recycling. Here, we describe how TC10 knockdown and TC10 functional mutants reduce AMPAR surface levels and synaptic currents by altering AMPAR recycling. TC10 mutants do not alter the increases in AMPAR surface levels that occur during LTP and only partially alter decreases in AMPAR surface levels and synaptic currents that occur during LTD. Overall, our findings indicate that TC10 mutants have differential effects on constitutive and activity-dependent AMPAR recycling

because AMPARs traffic through different endocytosis pathways, and activity-dependent events alter the endocytosis pathway taken by AMPARs. Furthermore, our findings of increased AMPAR endocytosis with LTP are inconsistent with the assumption that AMPAR recycling occurs locally at synapses. Instead, AMPARs added to LTP-stimulated synapses may be trafficked into these synapses from outside the local synaptic pool.

## Results

### Disrupting TC10 expression or function reduced the number of AMPARs at the cell surface

We first examined whether the small GTPase, TC10, had a role in AMPAR trafficking by knocking down TC10 expression (*Figure 1*). Knockdown was achieved using a short hairpin RNA construct (shRNA) that expressed a GFP reporter to identify neurons with the shRNA. Using real-time PCR, we observed that endogenous TC10 mRNA was reduced by 90% in cortical neurons (*Figure 1—figure supplement 1*). Neurons were co-transfected with AMPAR subunits, GluA1, with a fluorescent mCherry tag at the extracellular N-terminus (mCherry-GluA1) to simultaneously monitor GluA1 surface fluorescence and total GluA1 mCherry fluorescence. We found that the ratio of cell surface to total mCherry-GluA1 was similar from neuron to neuron (SEM = 14%). With the TC10 knockdown, the surface/total mCherry-GluA1 ratio for neurons was significantly reduced by 79% (*Figure 1A,D*) compared to control neurons that expressed GFP without the shRNA.

The surface levels of GluA1-containing AMPARs at somata were also reduced when we expressed TC10 mutants in the neurons. The T31N mutation or the 'dominant-negative' mutant (TC10DN) keeps TC10 in its inactive, GDP-bound state. The Q75L mutation or the 'constitutively active' mutant (TC10CA) is kept in its GTP-bound state (*Neudauer et al., 1998*). Expression of wild-type TC10 (TC10WT) did not significantly alter surface levels at somata assayed by mCherry-GluA1 transfection, whereas TC10DN and TC10CA mutants reduced GluA1-containing AMPAR surface levels by ∼50% (*Figure 1A,B*; statistical details in the legend; images of the corresponding whole neurons are displayed in *Figure 1—figure supplement 2*). While TC10 knockdown with shRNA did not alter somata morphology, dendritic morphology was altered and synapse numbers reduced (not shown). In contrast, TC10DN, TC10CA mutants, and TC10WT did not alter dendrite morphology (not shown) or synapse number (*Figure 1—figure supplement 3*). In dendrites, GluA1 surface levels were again reduced ∼50% by TC10DN and TC10CA and a small but significant increase of GluA1 surface levels (33% ± 10%) was observed with TC10WT expression (*Figure 1A*). We obtained similar results in dendrites with TC10WT, TC10DN, and TC10CA, when a GluA1-specific antibody that recognizes an extracellular epitope was used to quantify the endogenous surface AMPAR levels (*Figure 1F*).

We also performed paired whole cell recordings from synaptically coupled cultured hippocampal neurons to assay levels of functional AMPARs at the synapses. TC10DN and TC10CA reduced the synaptic AMPAR EPSC amplitudes by ∼60% (*Figure 1G,H*). Thus, the number of functional AMPARs at synapses was reduced to approximately the same extent as the total number of AMPARs on the cell surface as measured by immunostaining. Our findings with immunofluorescence and electrophysiology that expression of TC10DN and TC10CA in neurons caused the same effects on AMPAR levels in neurons are consistent with previous studies characterizing the role of TC10 in the secretory pathway. Assaying depolarization-induced secretion of neuropeptide Y (NPY) in PC12 cells, both TC10DN and TC10CA reduced secretion in the range of 40–60% (*Kawase et al., 2006*) similar to our findings with AMPARs (*Figure 1*). Using a TC10 FRET sensor construct to assay whether TC10 is in the GTP-TC10 or GDP-TC10 state, they concluded that the TC10 GTPase hydrolysis cycle is required for NPY secretion. TC10DN and TC10CA both reduced secretion by blocking the GTP hydrolysis cycle at different steps. Similar results were obtained assaying nerve growth factor-induced neurite outgrowth in PC12 cells (*Fujita et al., 2013*). Both papers suggested that during exocytosis TC10CA allowed cargo to load into transport vesicles to be delivered to target membranes. TC10CA blocked exocytosis by preventing the GTP-TC10 to GDP-TC10 transition required for transport vesicle docking and/or fusion. In contrast, the results suggest that TC10DN blocked the GDP-TC10 to GTP-TC10 transition, which blocked a different step, cargo loading onto vesicles thus preventing vesicle delivery to target membranes.

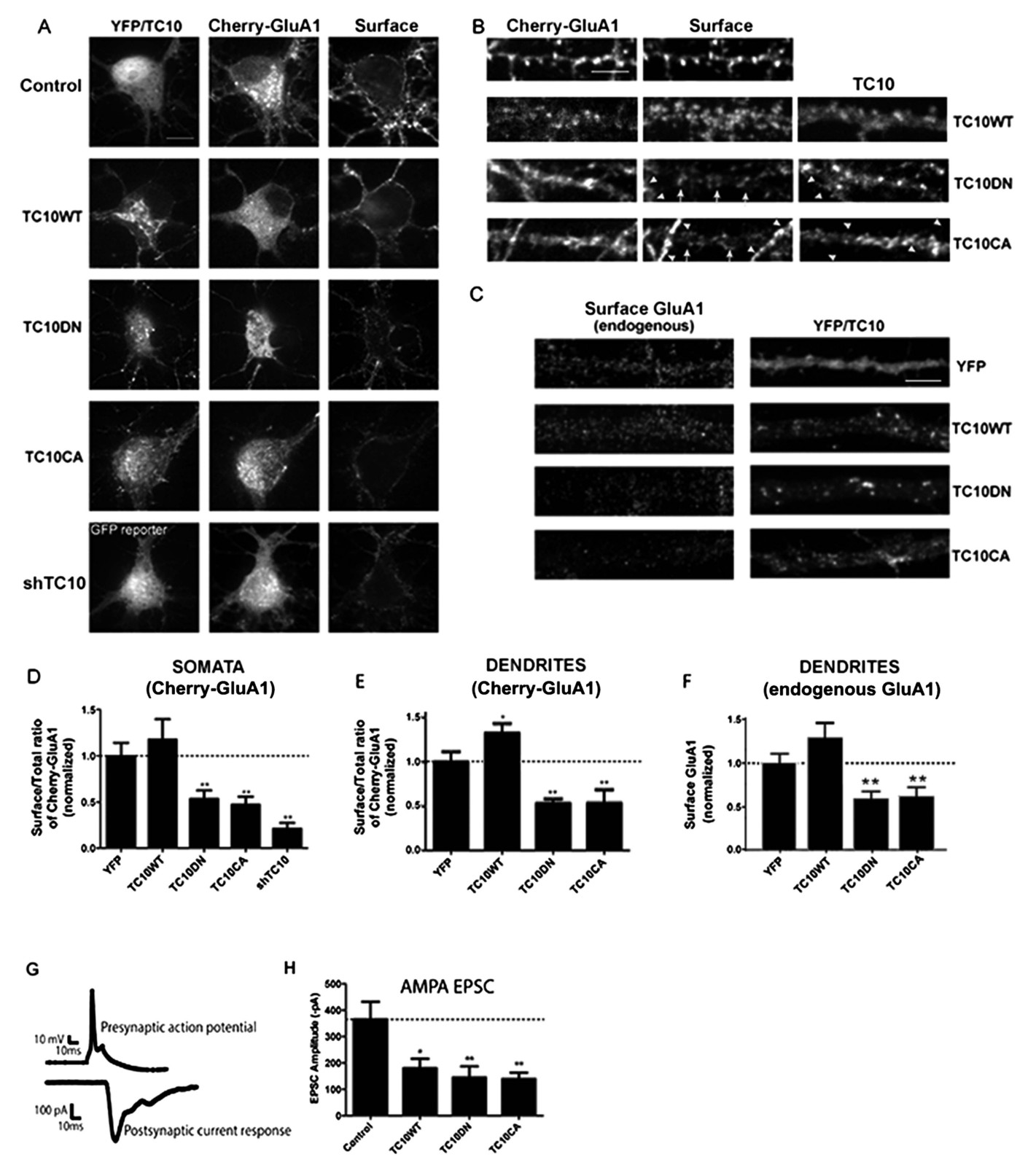

**Figure 1**. Disrupting TC10 level or function reduced AMPARs on the cell surface. (**A**) Representative somata of cultured rat hippocampal neurons (E18) expressing mCherry-tagged GluA1 subunits, free Venus (Control), shTC10RNA or Venus-tagged TC10WT, TC10DN or TC10CA. Intact, live neurons (DIV18) were stained with anti-RFP antibody to visualize surface AMPA-type glutamate receptors (AMPARs) and then fixed and processed. Images of the whole neuron for each of the shown somata are displayed in *Figure 1—figure supplement 2*. (Scale bar = 10 µm). (**B**) Representative dendrites of neurons

*Figure 1. continued on next page*

*Figure 1. Continued*

expressing mCherry-GluA1 without TC10 or with TC10WT, TC10DN, or TC10CA. Arrows indicate dendrites expressing TC10 mutants (TC10DN or CA) with weak surface expression of GluA1; arrowheads mark dendrites expressing mCherry-GluA1 without TC10 mutant expression and normal GluA1 surface expression level. (Scale bar = 5 μm). (**C**) Representative dendrites of intact neurons expressing either free Venus (Control), TC10WT, TC10DN, or TC10CA and labeled with an N-terminal, anti-GluA1 mAb to visualize endogenous, surface GluA1 receptors. (Scale bar = 5 μm). (**D**) Quantification of the surface/total ratio of mCherry-GluA1 at the somata of transfected cells in (**A**). Data are shown as ± SEM; control cells (YFP) 100% ± 14%, n = 32; TC10WT cells 118% ± 22%, n = 17; TC10DN cells 53% ± 9%, n = 23; TC10CA cells 47% ± 9%, n = 14; and shTC10 cells 21% ± 7%, n = 10. (**p < 0.03 relative to TC10WT). (**E**) Quantification of the surface/total ratio of mCherry-GluA1 in dendrites of transfected cells in (**B**). Data are shown as ± SEM; control cells (YFP) 100% ± 12%, n = 23; TC10WT cells 133% ± 11%%, n = 23; TC10DN cells 53% ± 5%, n = 23; and TC10CA cells 54% ± 15%, n = 8. (*p < 0.03 relative to YFP;**p < 0.02 relative to TC10WT). (**F**) Quantification of endogenous surface GluA1 on dendrites in (**C**). Data are shown as mean ± SEM; control cells (YFP) 100% ± 11%; TC10WT cells 130% ± 17%; TC10DN cells 59% ± 8%; and TC10CA cells 62% ± 10% (n = 7–10 cells per group; **p < 0.003 relative to TC10WT). (**G**) Representative whole-cell patch-clamp paired recordings showing the evoked presynaptic action potential (top) and the post-synaptic AMPAR-mediated current response (below). (**H**) Average AMPAR-mediated EPSC amplitudes of untransfected neurons (control) 365.2 ± 67 pA, n = 16 and neurons transfected with TC10WT 180.3 ± 36 pA, n = 8; TC10DN 144.5 ± 42 pA, n = 14; or TC10CA 139.2 ± 24 pA, n = 11. (*p < 0.04 relative to control;**p < 0.02 relative to control).

The following figure supplements are available for figure 1:

**Figure supplement 1**. TC10 RNAi (shRNA) knocked down endogenous TC10 expression in neurons.

**Figure supplement 2**. Low-magnification images of cultured rat hippocampal neurons (E18) corresponding to the somata shown in *Figure 1A*.

**Figure supplement 3**. TC10 mutants do not change synaptic density.

**Figure supplement 4**. TC10 mutants do not change the expression level of GluA1.

**Figure supplement 5**. A comparison of somatic ER and the distribution of exogenous mCherry-GluA1 subunits and endogenous GluA1 subunits.

**Figure supplement 6**. The effects of TC10 mutants on AMPARs in the somatic Golgi and in dendritic shafts.

Unexpectedly, in the paired whole-cell recordings from synaptically coupled cultured hippocampal neurons, TC10WT reduced synaptic currents by 51% (*Figure 1E,F*) even though TC10WT increased cell-surface AMPARs by 33% (*Figure 1B,C*). This differential effect of TC10WT suggests that TC10 function somehow distinguishes between AMPAR trafficking to and/or from synaptic sites compared to other sites on the cell surface of dendrites.

## TC10 regulates AMPAR trafficking through an Arf6-containing endocytosis pathway in dendrites

To explore how perturbing TC10 function reduced AMPAR surface levels, we examined whether TC10 mutants altered endogenous GluA1 subunit levels. Lentiviral infection of ~90% of the cultured neurons with TC10WT or the TC10 mutants did not alter endogenous GluA1 subunit levels, indicating that TC10 mutants do not alter AMPAR subunit synthesis or degradation (*Figure 1—figure supplement 4*). Nor did the TC10 mutant expression appear to alter AMPAR trafficking through the secretory pathway in the soma (*Figure 1—figure supplement 5*). Previously, we had found that AMPAR loss in dendrites correlated with AMPAR accumulation in the Golgi, which blocked AMPAR transport from somata to dendrites of cultured neurons (*Jeyifous et al., 2009*). Consistent with this possibility, TC10WT, TC10DN, and TC10CA all strongly co-localized with the Golgi marker GM130 in somata (*Figure 1—figure supplement 6A*). However, little mCherry-GluA1 was found in the Golgi, and most GluA1 subunits co-localized with endoplasmic reticulum (ER) markers (*Figure 1—figure supplement 5*), as previously observed for newly synthesized AMPAR subunits (*Greger et al., 2002*). We observed a similar distribution for the native GluA1 subunits in the ER (*Figure 1—figure supplement 5*) indicating that heterologous express of mCherry-GluA1 did not greatly increase levels of GluA1 in the ER. The small amount of mCherry-GluA1 that co-localized with the Golgi marker did not significantly change with TC10DN and TC10CA expression compared to that of TC10WT (*Figure 1—figure supplement 6A,B*). In fact, when TC10 mutants were expressed, there was a significant increase in mCherry-GluA1

levels in the shafts of dendrites (*Figure 1—figure supplement 6C*), suggesting that the TC10 mutants cause an accumulation of AMPAR intracellular levels in the dendritic shafts, but not in the somata, that results in the decreases in surface levels.

We next examined whether the TC10 mutants caused AMPARs to accumulate during their recycling in dendritic shafts. It is well established that AMPARs enter REs in dendritic shafts after synaptic stimulation with AMPA or NMDA (*Malinow and Malenka, 2002*; *Newpher and Ehlers, 2008*). To test for accumulation of AMPARs in REs, we assayed whether TC10 mutants increased endogenous AMPAR co-localization with TfRs, which largely exist in REs in dendritic shafts where they co-localize with GluA1-containing AMPARs (*Carroll et al., 1999*; *Ehlers, 2000*). Surprisingly, there was little co-localization, (4–6%), between endogenous GluA1 and TfR in dendrites under these conditions. The percentage of co-localization did not change for non-transfected neurons or for neurons expressing Venus-tagged TC10WT, TC10DN, or TC10CA (*Figure 2A,B*). TC10WT, TC10DN, or TC10CA expression also did not alter the distribution of TfRs in the dendrites and did not significantly affect the recycling of TfRs (data not shown). To confirm the validity of TfRs as a recycling endosome marker, we tested another recycling endosome protein, Rab11, and found significant co-localization between mCherry-tagged Rab11 and TfRs confirming that TfRs are largely in REs in dendrites (*Figure 2—figure supplement 1*).

Since few GluA1-containing AMPARs accumulated in TfR-labeled REs, it is possible that AMPAR recycling occurs via a different endocyctosis pathway not taken by TfRs. Several other endocytosis pathways exist (*Doherty and McMahon, 2009*). In particular, one endocyctosis pathway that utilizes the small GTPase, Arf6, occurs in dendrites (*Gong et al., 2007*; *Lavezzari and Roche, 2007*) and may be involved in AMPAR endocytosis (*Scholz et al., 2010*). In contrast to the ~5% co-localization observed between GluA1 and TfR puncta, we observed a significantly higher degree of overlap (Pearson's correlation coefficient, PCC = 0.46) between endogenous GluA1 subunits and transfected HA-Arf6 (*Figure 2C,D*), an established marker of endosomes in the Arf6, clathrin-independent pathway (*Donaldson et al., 2009*). Importantly, there was a high degree of co-localization between TC10WT and HA-Arf6 (PCC = 0.77), indicating that TC10 is largely found in Arf6 endosomes in dendrites (*Figure 2E*).

Consistent with TC10 having a role in AMPAR recycling through Arf6 endosomes, the TC10 mutants caused significant changes in the distribution of AMPAR intracellular puncta that co-localized with Arf6 (*Figure 2C,D*). TC10DN significantly increased GluA1 co-localization with Arf6, changing from a PCC of 0.46 to 0.66 (*Figure 2C,D*). Increases in AMPARs co-localizing with Arf6 appeared to be at subdomains within Arf6 endosomes (*Figure 2C*), consistent with TC10CA blocking AMPAR exit, and their accumulation in Arf6-endosomes. The TC10CA mutant had the opposite effect. TC10CA caused GluA1 co-localization with Arf6 to significantly decrease from a PCC of 0.46 to 0.31, consistent with increased AMPAR exit from Arf6 endosomes (*Figure 2C,D*). Because TC10CA caused a ~50% decrease in surface AMPAR, similar to the effects of TC10DN (*Figure 1B,C and F*), increases in AMPARs exiting from Arf6 endosomes were not inserted into the cell surface. Expression of TC10CA changed the size and number of the GluA1-containing puncta in dendritic shafts. GluA1-containing puncta were much more numerous compared to TC10WT, with an increased number of smaller puncta and a loss of most of the larger puncta in dendrites (*Figure 2E*). This result suggests that the AMPARs exiting from Arf6 endosomes remained in smaller transport vesicles because TC10CA blocked their exocytosis at the plasma membrane. These interpretations of the effects of the TC10DN and TC10CA mutants are supported by the conclusions of previous studies of the effects of the TC10 mutants on the distribution of different cargo during secretion (*Kawase et al., 2006*; *Fujita et al., 2013*); specifically that TC10DN blocked the GDP-TC10 to GTP-TC10 transition and cargo loading onto the transport vesicle, while TC10CA blocked the GTP-TC10 to GDP-TC10 transition and the transport vesicle exocytosis.

## Constitutive AMPAR endocytosis is independent of dynamin

The clathrin-dependent endocytosis pathway taken by TfRs requires dynamin function, while Arf6-dependent clathrin-independent endocytosis is independent of dynamin function (*Doherty and McMahon, 2009*). As another test of the AMPAR endocytosis pathway, we used the reagent, dynasore, to block dynamin activity in neurons and assayed TfR and AMPAR endocytosis. In these experiments, we performed two different sets of assays to quantitatively measure how dynasore affected AMPAR and TfR internalization. In the first assay, mCherry-tagged GluA1 was expressed as in *Figure 1*, and an 'antibody feeding' assay was used to label only the internalized mCherry-tagged,

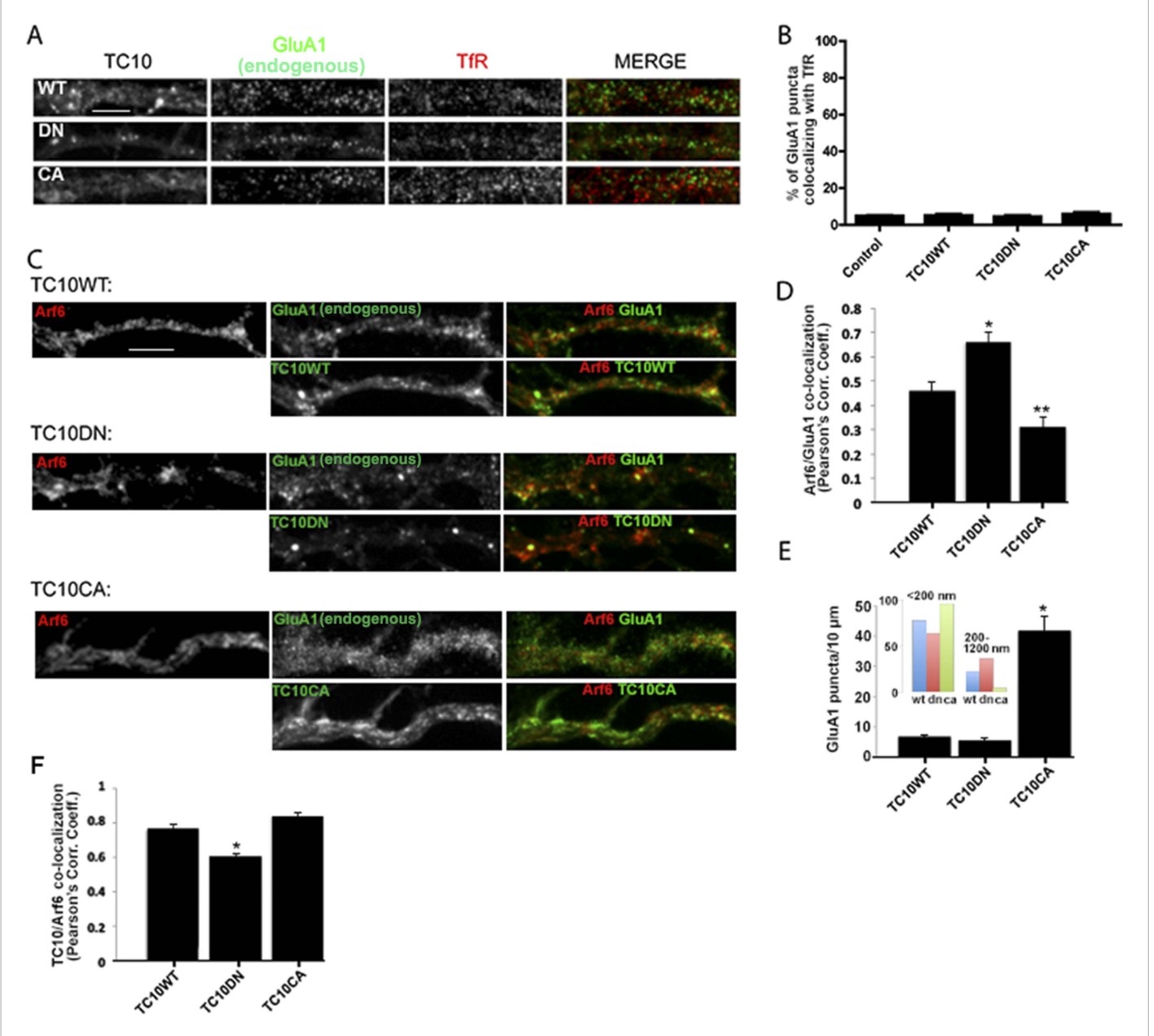

**Figure 2**. TC10 regulates AMPAR trafficking through an Arf6-containing endocytosis pathway in dendrites. (**A**) Effects of TC10 constructs on the co-localization between endogenous GluA1 subunits and transferrin receptors (TfRs). Cultured neurons were transfected with Venus-tagged TC10WT or TC10DN/CA mutants and permeabilized cells stained for total GluA1 and TfR (Tf-Alexa 647). GluA1 showed little co-localization with Tf-labeled endosomes in dendrites. (Scale bar = 5 μm). (**B**) Quantification of GluA1 and TfR co-localization. Images in A were analyzed to measure the percent of GluA1 puncta co-localizing with TfR puncta. Expression of TC10 constructs did not alter the degree of co-localization. Data are shown as mean ± SEM; control cells 4.6% ± 0.3%; TC10WT cells 4.9% ± 0.8%; TC10DN cells 4.2% ± 0.8%; TC10CA cells 5.6% ± 1%, n = 5 for all groups. (**C**) Effects of TC10 constructs on the co-localization between GluA1 subunits and Arf6. Neurons were transfected with Arf6-HA and Venus-TC10WT or TC10DN/CA mutants. Cells were permeabilized and stained for Arf6-HA (Rb anti-HA) and total GluA1 (anti-GluA1 mAb). (Scale bar = 5 μm). (**D**) Quantification of the overlap between GluA1 and Arf6. Images in C were analyzed to measure the Pearson's correlation coefficients ($R_r$) of GluA1 co-localization with Arf6. Data are shown as mean ± SEM; TC10WT 0.46 ± 0.04; TC10DN 0.66 ± 0.04; TC10CA 0.31 ± 0.04 (n = 7-10 cells per group; *p < 0.02 relative to TC10WT; **p < 0.05 relative to TC10WT). (**E**) Effects of TC10 constructs on GluA1 puncta density (number of puncta per 10 μm) in dendrites of neurons expressing Venus-TC10WT or TC10DN/CA mutants. Data are shown as mean ± SEM; TC10WT 6.9 ± 0.4; TC10DN 5.6 ± 0.7; TC10CA 41.7 ± 4.8 (n = 5 fields per group; *p < 0.0002 relative to TC10WT). Inset, histogram showing the distribution of small (diameter <200 nm) and large (diameter 200–1200 nm) GluA1 puncta
*Figure 2. continued on next page*

*Figure 2. Continued*

(n = ~100 puncta per group). (**F**) Quantification of the overlap (Pearson's correlation coefficients, $R_r$) between TC10 and Arf6 in (**C**). Data are shown as mean ± SEM; TC10WT 0.77 ± 0.03; TC10DN 0.60 ± 0.02; TC10CA 0.84 ± 0.02 (n = 7–10 cells per group; *p < 0.0004 relative to TC10WT).

The following figure supplement is available for figure 2:

**Figure supplement 1**. Co-distribution of TfR staining and mCherry-Rab11.

GluA1 AMPARs, and TfRs (*Figure 3A,B*). After dynasore treatment, we compared the distribution of internalized mCherry-GluA1 subunits to that of TfRs surface labeled with Alexa Fluor 647-conjugated transferrin (Tf-Alexa 647; *Figure 3A*). Consistent with a block of its endocytosis, dynasore treatment appeared to cause TfRs to buildup at the plasma membrane (*Figure 3A*). Tf-Alexa 647 uptake was inhibited by 70% ± 18% in the presence of dynasore (*Figure 3B,C*). However, dynasore treatment did not significantly alter the levels (*Figure 3C*) or distribution (*Figure 3A*) of internalized mCherry-GluA1 subunits.

In the second assay, endogenous AMPAR internalization was assayed by biotinylating cell-surface proteins with a cleavable biotinylation cross-linking reagent and treating intact neurons with glutathione, a membrane impermeable reagent after internalization. Dynasore treatment reduced the levels of internalized TfR by 61% ± 6% (*Figure 3D,E*), consistent with endocytosis predominantly through clathrin-dependent endocytosis pathway. In contrast, dynasore treatment reduced internalized GluA1 subunits by only 10% ± 9% (*Figure 3D,E*), consistent with endocytosis predominantly through a different endocytosis pathway. Based on these results together with previous data (*Figure 2*), we conclude that TC10 has a role in AMPAR recycling through an Arf6-dependent recycling pathway that is different from the TfR recycling pathway.

## Changes in synaptic activity alter the recycling pathway taken by AMPARs

Evidence that AMPARs are in REs and clathrin-coated pits comes almost exclusively from studies examining AMPAR activity-dependent recycling. Our findings of AMPAR recycling via a TC10- and Arf6-dependent, dynamin-independent pathway occurred during constitutive AMPAR recycling. We, therefore, investigated whether the AMPAR endocytosis pathway changes under conditions where activity-dependent recycling occurs. First, we specifically tested how expressing TC10DN or TC10CA in neurons altered LTP and LTD. Chemically inducing LTP (cLTP) with glycine stimulation to activate synaptic NMDARs increased the AMPAR surface/total ratio by 39% ± 10% (p < 0.1, n = 10, 14), while chemically inducing LTD (cLTD) with acute NMDA treatment, decreased the AMPAR surface/total ratio by 62% ± 3% (*Figure 4A,B*).

Neurons expressing TC10DN or TC10CA showed increased AMPAR surface/total ratios with cLTP (*Figure 4A*). The percent increase in surface AMPARs was even larger than controls (*Figure 4B*). The larger percent increase may be explained by the fact that the cLTP-induced increases in surface AMPARs were about the same size as controls, while the levels of surface AMPARs were reduced by TC10 variants (*Figure 1*). The finding that TC10DN or TC10CA expression had no negative effect on cLTP-induced increases but reduced surface AMPAR levels prior to cLTP is again consistent with AMPAR recycling occurring through multiple endocytosis pathways, one pathway providing AMPARs inserted during cLTP and the other pathway primarily involved in constitutive AMPAR recycling.

In neurons expressing TC10 mutants during cLTD, we observed only half the cLTD-induced decrease in AMPARs with TC10DN, while TC10CA did not alter the cLTD-induced decrease (*Figure 4A,B*). These results are different from what occurs during LTP and suggest that the TC10- and Arf6-dependent endocytosis pathway is involved to some degree in the AMPAR trafficking during LTD that removes AMPARs from synapses. To more directly test how TC10DN and TC10CA affect the trafficking of synaptic AMPARs during LTD, we induced LTD by low-frequency synaptic stimulation (*Montgomery and Madison, 2002*; *Montgomery et al., 2005*) (*Figure 4C*). Similar to what we observed with cLTD, the AMPAR-mediated synaptic currents were reduced by 52.4% ± 0.5% by LTD. During LTD, AMPAR EPSCs were reduced by 27.5% ± 1.0% in the presence of

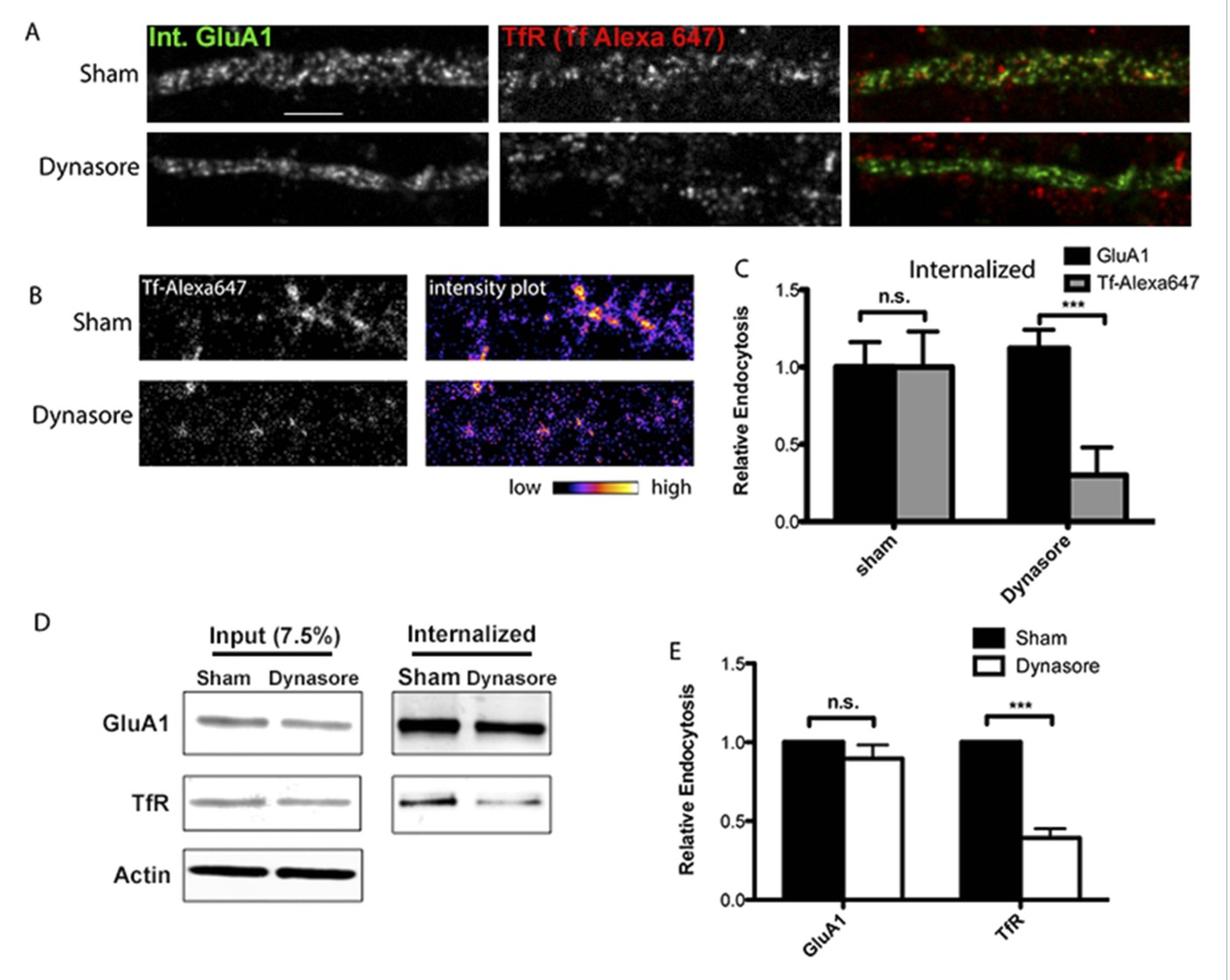

**Figure 3**. AMPARs undergo dynamin-independent endocytosis. (**A**) mCherry-GluA1 and TfR internalization and block of dynamin function. Hippocampal neurons were transfected with mCherry-GluA1, and 1 day post-transfection, treated with 1% DMSO (control group) or 80 µM dynasore (to block dynamin function) for 30 min at 37°C. Neurons were then incubated with anti-RFP antibodies and Tf-Alexa 647 at 37°C for 40 min, acid washed to remove surface antibodies/dye, fixed, permeabilized, and imaged. With sham treatment (control), surface-labeled GluA1 and TfR were both internalized in the dendritic shaft (first row). Dynasore treatment had no effect on GluA1 internalization, but blocked TfR endocytosis (Tf-Alexa 647 labeling confined to surface and not present intracellularly, second row). (**B**) Intensity plots of internalized TfR (Tf-Alexa 647) with and without (sham) dynasore treatment as in (**A**). Dynasore greatly inhibited TfR internalization. (**C**) Quantification of GluA1 and Tf-Alexa647 internalization and block of dynamin function. Data are shown as mean ± SEM; GluA1 endocytosis, sham 100% ± 16%; dynasore 112% ± 12%. For Tf-Alexa647 endocytosis, sham 100% ± 23%; dynasore 30% ± 18%, (n = 5 for all groups; ***p < 0.001). (**D**) Endogenous GluA1 and TfR internalization and block of dynamin function. As an alternative approach to measure GluA1 and TfR Internalization, cortical neurons were sham treated (with DMSO) or with dynasore as in (**A**). After, surface proteins were labeled with Sulfo-NHS-SS-biotin and cultured for 40 min at 37°C to allow for endocytosis. Biotin on proteins remaining on the cell surface was removed by glutathione treatment and cells solubilized. Internalized proteins were pulled down with streptavidin beads and analyzed by Western blotting. Displayed are GluA1, TrR, and actin bands (loading control) from whole cell lysates to estimate inputs for sham- or dynasore-treated neurons (left) and GluA1 and TfR bands from the streptavidin pull-downs to estimate internalized receptors for sham- or dynasore-treated neurons (right). (**E**) Quantification of GluA1 and TfR internalization and block of dynamin function. The levels of GluA1 endocytosis were reduced to 90% ± 9% vs sham treated by dynasore treatment. The levels of TfR endocytosis were reduced to 61% ± 6% (n = 3 experiments; ***p < 0.01) vs sham treated by dynasore treatment. Total protein levels (inputs) for GluA1 and TfR were not affected by dynasore treatment.

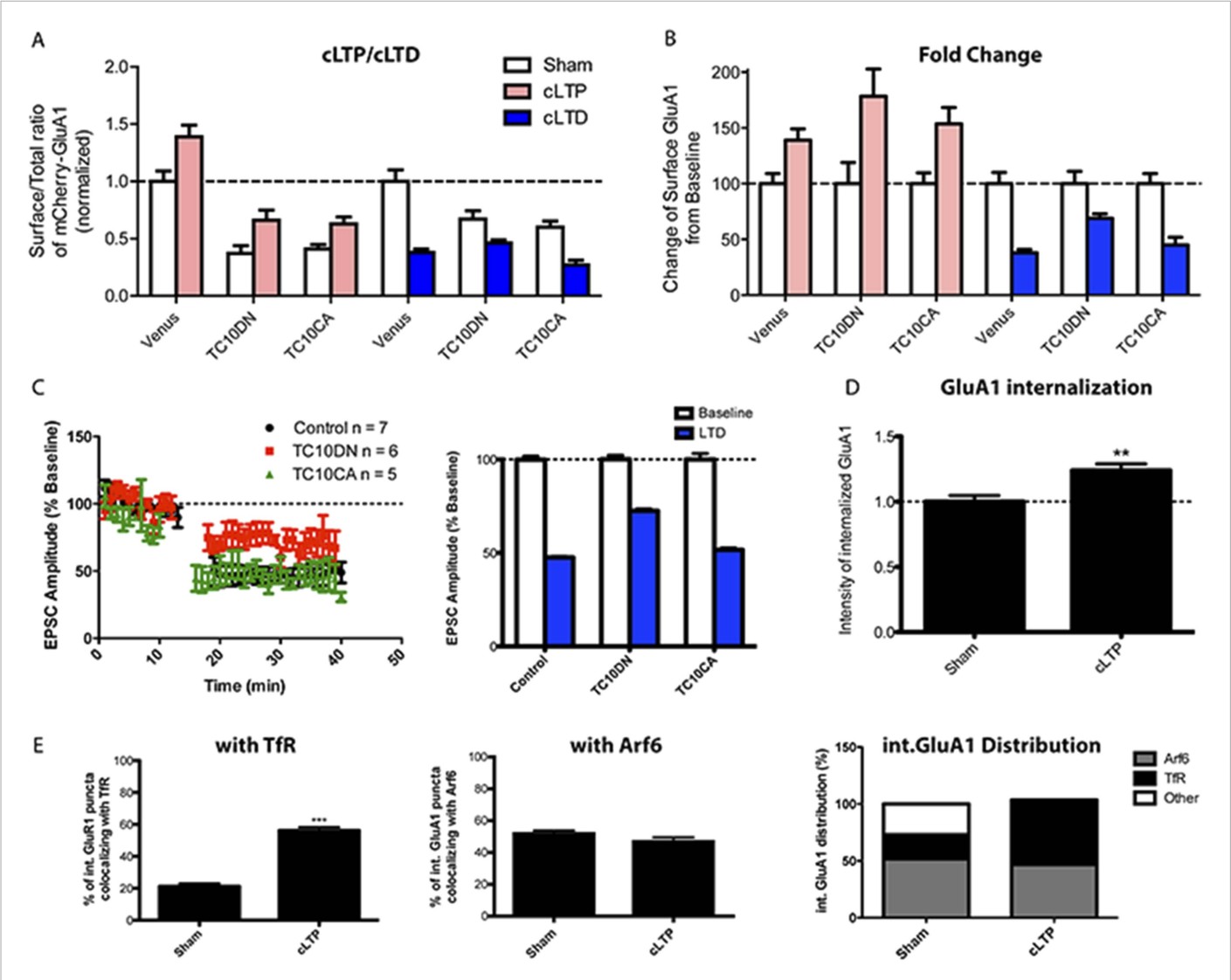

**Figure 4**. Synaptic activity alters the endocytosis pathway taken by AMPARs. (**A**) Effects of TC10 constructs on chemically inducing LTP (cLTP) and chemically inducing LTD (cLTD). Hippocampal neurons were transfected with mCherry-GluA1 plus Venus, TC10DN, or TC10CA. 1 day post-transfection, cells were stimulated with a mixture of the NMDA receptor agonist, glycine, and $GABA_A$ receptors antagonists to induce long-term potentiation (LTP) chemically in hippocampal cultures. Then, surface exposed mCherry-GluA1 was stained live with anti-RFP antibody. In control cells, cLTP treatment caused a 39% increase in surface GluA1 (100% ± 9%, n = 10–139% ± 10%, n = 14). In cells expressing TC10DN and TC10CA, cLTP treatment also caused a 78% (37% ± 7%, n = 13–66% ± 9%, n = 12) and 54% (41% ± 4%, n = 21–63% ± 6%, n = 15) increase in surface GluA1, respectively. In parallel, transfected cells were treated with NMDA to chemically induce long-term depression (LTD) as well. Control cells showed a 62% decrease of surface GluA1 (100% ± 10%, n = 10–38% ± 3%, n = 11), while TC10DN and TC10CA expressing cells showed a 31% (67% ± 7%, n = 13–46% ± 3%, n = 10) and 55% (60% ± 5%, n = 5–27% ± 4.2%, n = 6) reduction, respectively. (**B**) Effects of TC10 constructs on cLTP and cLTD. Data in (**A**) replotted and normalized to respective sham values in order to compare the changes with cLTP and cLTD. (**C**) Effects of TC10 constructs on synaptically induced LTD. Left: AMPAR-EPSC amplitudes, expressed as a percentage of baseline AMPAR-EPSC amplitude (10 min), before and after the induction of LTD by electrical LFS for 5 min (gap). Right: Bar graph of the average AMPAR-EPSC amplitude depression measured 20 min after LFS. Control cells showed a 52.4% ± 0.5% reduction in AMPAR-EPSC amplitude after LTD induction; TC10DN and TC10CA showed a 27.5% ± 1.0% and 48.3% ± 1.0% reduction in AMPAR-EPSC amplitude, respectively. (**D**) mCherry-GluA1 internalization after cLTP. Neurons were sham or cLTP treated and imaged for internalized mCherry-GluA1 and either TfR or HA-Arf6. GluA1 internalization increased by 24% ± 5% when treated with cLTP (n = 6; **p < 0.01). (**E**) mCherry-GluA1 co-localization with TfR (left) and Arf6 (middle) after cLTP. Co-localization of internalized GluA1 with TfR increased with cLTP: sham-treated 21% ± 2%, n = 15; cLTP 56% ± 2%, n = 23 (***p < 0.0001). Co-localization of internalized GluA1 with Arf6 did not significantly change: sham treated 52% ± 2%, n = 12; cLTP 47% ± 3%, n = 5. Left panel displays the distribution of GluA1 co-localization with TfR, Arf6, or neither marker for sham treated and cLTP conditions.

The following figure supplement is available for figure 4:

**Figure supplement 1**. Synaptic activity alters the endocytosis pathway taken by AMPARs.

TC10DN, and TC10CA had no effect on LTD (a 48.3% ± 1.0% reduction) (*Figure 4C*). We could not perform similar experiments with LTP because LTP cannot be induced by electrical stimulation in dissociated cultures (*Shi et al., 1999*; *Li et al., 2011*). However in a previous study, we found that the same cLTP protocol applied to dissociated hippocampal neuronal cultures caused a LTP of EPSC amplitudes (*Li et al., 2011*).

Our results with TC10DN or TC10CA expression on AMPAR trafficking during cLTP suggest that constitutive and activity-dependent AMPAR recycling occurs through different pathways. To further test this possibility, we assayed how cLTP altered the levels of internalized GluA1 subunit co-localization with TfRs or with Arf6-HA. We observed that 21% ± 2% of the internalized AMPARs co-localized with TfR (*Figure 4D,E*), consistent with a previous estimate of the amount of internalized AMPARs that co-localized with TfR during constitutive recycling (*Park et al., 2004*). 52% of the internalized AMPAR co-localized with Arf6-HA and 27% of internalized AMPARs did not co-localize with TfRs or with Arf6-HA (*Figure 4E*). This pool of internalized AMPARs is likely in a different endosomal pool such as early endosomes, late endosomes, or lysosomes (*Fernández-Monreal et al., 2012*). After cLTP induction, the distribution of internalized AMPARs in the different pools changed: cLTP increased the amount of internalized GluA1 that co-localized with TfR from 21% to 56% (*Figure 4E*; *Figure 4—figure supplement 1A*), while the internalized GluA1 that co-localized with Arf6 was unchanged (52% vs 47%; *Figure 4E*; *Figure 4—figure supplement 1B*). cLTP increased the total amount of internalized GluA1 by 24% ± 5% (*Figure 4D*), while the pool of internalized AMPARs that did not co-localize with TfRs or with Arf6-HA was no longer measurable. Our results are consistent with significant changes in AMPAR endocytosis and recycling that occurs during cLTP. During constitutive recycling, AMPARs appear to be recycling primarily through a dynamin-independent, Arf6-, and TC10-dependent pathway though some recycle through a dynamin-dependent, TfR-containing pathway. After cLTP, a much larger fraction of the AMPARs recycle through the dynamin-dependent pathway taken by TfRs. Our results further suggest that after LTP, fewer AMPARs are trafficked for degradation via a late endosomal/lysosomal pathway as previously described (*Fernández-Monreal et al., 2012*).

## Discussion

### Constitutive AMPAR recycling and TC10

The findings of this study are consistent with constitutive AMPAR recycling occurring largely through a pathway different from the clathrin-dependent pathway used by TfRs. The data in support of an alternative recycling pathway are that internalized AMPARs co-localize much more with Arf6 than with TfRs (*Figure 2*), and that inhibition of dynamin function blocks TfR endocytosis without significantly altering AMPAR endocytosis (*Figure 3*). The features of the alternative AMPAR recycling pathway, for example, presence of Arf6 and independence from dynamin function, are consistent with a clathrin-independent endocytosis pathway (*Doherty and McMahon, 2009*; *Donaldson et al., 2009*). The Arf6-mediated, clathrin-independent pathway is a separate endocytosis pathway that exists in neurons and mediates the endocytosis and recycling of a number of different receptors and transport proteins. These include the metabotropic glutamate receptors (mGluR), mGluR5 (*Fourgeaud et al., 2003*) and mGluR7 (*Lavezzari and Roche, 2007*), and the potassium channel Kir3.4 (*Gong et al., 2007*). AMPARs were shown to undergo a clathrin-independent recycling pathway, in addition to clathrin-mediated pathway in *Caenorhabditis elegans* (*Glodowski et al., 2007*). Another study using electron microscopy (EM) (*Tao-Cheng et al., 2011*) also suggests that constitutive AMPAR recycling occurs through a different pathway. The study found that all intracellular structures with the features of REs were labeled for TfRs in dendritic shafts of cultured rat hippocampal neurons but only 28% of these REs were labeled for AMPARs. If AMPAR endocytosis occurs through a single, clathrin-dependent pathway, AMPARs would all enter clathrin-coated pits during constitutive AMPAR recycling. Entry into clathrin-coated pits can only be unambiguously resolved at the EM level. EM studies assaying AMPAR subunit localization in clathrin-coated pits observed few AMPARs in clathrin-coated pits (*Petralia et al., 2003*; *Tao-Cheng et al., 2011*). Tao-Cheng et al. found that 76% of the clathrin-coated pits contained TfRs but only 24% of the pits contained AMPAR subunits. Clathrin-coated pits near PSDs have been proposed to be specialized endocytic zones (EZs) (*Blanpied et al., 2002*; *Racz et al., 2004*) that mediate endocytosis of AMPA receptors for local recycling in spines (*Lu et al., 2007*; *Petrini et al., 2009*; *Kennedy et al., 2010*). Both EM studies failed to detect any AMPAR labeling in the EZs at synapses

under conditions where constitutive AMPAR recycling was occurring (*Petralia et al., 2003*; *Tao-Cheng et al., 2011*).

In this study, we have also characterized in detail the role of the small Rho GTPase, TC10, in AMPAR recycling through the Arf6-mediated, clathrin-independent pathway. We found that altering TC10 expression and function in neurons reduced levels of cell-surface AMPARs. The TC10 mutants, TCDN and TC10CA, equally reduced cell-surface AMPARs by ~50% but did not significantly affect AMPAR trafficking through the secretory pathway. Normal levels of AMPARs departed from the somatic Golgi and were transported to dendrites and synapses. However, the TC10 mutants had differential effects on where AMPARs accumulated in dendritic shafts. TC10DN reduced surface AMPARs by causing increased AMPAR accumulation in Arf6 endosomes apparently by blocking their exit from the endosomes. TC10CA reduced surface AMPARs by increasing their exit from Arf6 endosomes and blocking their exocytosis, thereby increasing what appear to be AMPAR transport vesicles in the dendritic shafts.

Results from a previous study suggest that the associations between TC10 and AMPARs are indirect, requiring an adaptor protein, nPIST, which interacts with the AMPAR TARP subunit (*Cuadra et al., 2004*). nPIST, like TC10, primarily co-localizes with Golgi markers in the somata of cultured hippocampal neurons. But it is also found in puncta in dendritic shafts, not in spines, and the puncta do not co-localize with Golgi membranes (*Chen et al., 2012*). nPIST interactions with TC10 in dendrites, thus, are likely at the Arf6 endosomes where we observed most of TC10 in dendrites. One possibility is that TC10 acts to regulate interactions between nPIST and AMPAR TARP subunits when present together in Arf6 endosomes, and thereby, regulate the trafficking of AMPARs from Arf6 endosomes to dendritic exocytosis sites.

## Activity-dependent AMPAR recycling

Our findings that AMPARs recycle through two different pathways provide new insights into how AMPAR recycling is altered in response to changes in synaptic activity. The increase in AMPARs in REs after cLTP reflects a redistribution of trafficking AMPARs in dendrites such that AMPAR receptor recycling via REs is increased, while recycling via the Arf6-TC10-containing endosomes was unchanged. We also observed a third pool of endocytosed AMPARs that did not co-localize with either TfR or Arf6. This third pool, which should include AMPARs in early endosomes, late endosomes and lysosomes, decreased from 27% of the total to essentially 0%. A previous study suggested that during LTD an activity-dependent switch occurs such that more endocytosed AMPARs in early endosomes were routed to the Rab7-dependent pathway to lysosomes and less were routed to the recycling endosome pathway (*Fernández-Monreal et al., 2012*). Our data suggest that the opposite is occurring during cLTP. That is, an activity-dependent switch occurs such that few to none of the endocytosed AMPARs in early endosomes are routed via the Rab7-dependent pathway to lysosomes. Instead, virtually all AMPARs in early endosomes are routed to the REs. However, this switch can only explain part of the increase in AMPARs in REs during cLTP because the total number of endocytosed AMPARs increased by an additional 24%. Thus, part of the increase in AMPARs in REs during cLTP appears to be caused by an increase in AMPAR endocytosis, presumably at clathrin-coated pits

Expression of TC10DN increased AMPARs in Arf6-containing endosomes, while TC10CA decreased AMPARs in these endosomes (*Figure 2C,D*). The differential effects of TC10DN and TC10CA on AMPAR localization in Arf6-containing endosomes provide a potential explanation for the differential effects of TC10DN and TC10CA during LTD. It is possible that AMPARs excluded from Arf6-containing endosomes by TC10CA have access to the endosomal pathway used during LTD. The AMPARs added to the Arf6-containing endosomes with TC10DN expression do not have access to the endosomal pathway used during LTD.

Previous studies have found that synaptic stimulation increases dynamin- and clathrin-dependent AMPAR endocytosis (*Carroll et al., 1999*; *Lüscher et al., 1999*; *Ehlers, 2000*; *Lee et al., 2002*; *Scholz et al., 2010*; *Tao-Cheng et al., 2011*). In some of these studies, blocking dynamin/clathrin-dependent endocytosis prevented LTD but did not alter constitutive AMPAR endocytosis. It was also found that blocking constitutive AMPAR endocytosis by interfering with NSF binding to GluA2 subunits does not alter LTD or activity-dependent AMPAR endocytosis (*Carroll et al., 1999*; *Lüscher et al., 1999*; *Ehlers, 2000*; *Lee et al., 2002*). Altogether, these studies demonstrate that activity-dependent and constitutive AMPAR recycling can be uncoupled, and thus, appears to be independently regulated, consistent with separate processes underlying activity-dependent and constitutive AMPAR recycling.

Our results with TC10 mutants are also consistent with separate processes underlying activity-dependent and constitutive AMPAR recycling.

## Why are there different AMPAR endocytosis pathways?

The results from this study suggest that the two different AMPAR recycling pathways serve different functions. The Arf6-dependent recycling pathway, also dependent on TC10 function, predominates during AMPAR constitutive recycling. Another recycling pathway, which is dynamin and clathrin dependent, increases during AMPAR activity-dependent recycling (cLTP and cLTD). The simplest model of AMPAR recycling based on our data is displayed in *Figure 5B*. To explain the effects of dynamin inhibition, we propose that the two recycling pathways originate at separate endocytosis sites, either clathrin-coated pits or clathrin-independent sites. As proposed by others (*Donaldson et al., 2009*) but not shown in the model, the two recycling pathways merge at early endosomes, where AMPARs would be sorted for trafficking into the various pools, either the REs used by TfRs, the Arf6-depedent REs or late endosomes for lysosomal degradation. As proposed in the model, the two recycling pathways end when AMPARs are exocytosed at separate sites at the cell membrane. The presence of two different AMPAR recycling exocytosis sites may help to explain recent data describing different AMPAR recycling exocytosis sites (*Kennedy et al., 2010*; *Ahmad et al., 2012*). We envision that the Arf6/TC10-dependent recycling pathway has largely a caretaker role, delivering AMPARs to early endosomes where a decision is made to degrade internalized AMPARs or return them to the plasma membrane. The dynamin/clathrin-dependent recycling functions predominantly during synaptic activation and appear to have a different role than the Arf6/TC10-depedent recycling pathway (*Figure 5*). Other receptors are regulated in a similar way by the same two recycling pathways. β2-adrenergic and M3 muscarinic receptors undergo constitutive recycling when not activated by ligand via an Arf6-dependent, clathrin-independent pathway. After ligand activation, their recycling pathway switches and recycling occurs via the clathrin-dependent pathway (*Scarselli and Donaldson, 2009*). The recycling pathway of α1-integrin receptors also switches after their activation (*Arjonen et al., 2012*).

Our finding that AMPAR recycling through the dynamin/clathrin-dependent pathway increased during cLTP provides insights into the function of this pathway during AMPAR recycling. Previously, it was assumed that AMPAR recycling was largely confined to a single synaptic spine and the dendrite area nearby (*Figure 5A*). Increases in synaptic AMPARs during NMDAR-dependent LTP were thought to increase AMPAR exocytosis without increasing AMPAR endocytosis causing a decrease in the level of AMPARs in REs (*Malinow and Malenka, 2002*; *Bredt and Nicoll, 2003*; *Collingridge et al., 2004*; *Park et al., 2004*; *Shepherd and Huganir, 2007*). However, we found that AMPAR levels in REs increased after LTP, and endocytosis increased in parallel with increased AMPAR levels at synapses during cLTP. Our results suggest that AMPAR recycling is not limited to trafficking AMPARs into and out of the same synaptic spine. Instead, we suggest that AMPAR recycling has the additional function of transporting AMPARs to sites distant from where they originate. Increased AMPAR endocytosis during cLTP occurs through the dynamin-dependent recycling pathway starting at clathrin-coated pits at sites distant from synapses undergoing cLTP (*Figure 5C*). Consistent with this idea, AMPARs were not found in clathrin-coated pits at spines but in clathrin-coated pits well outside spines along the dendritic shafts (*Tao-Cheng et al., 2011*). After endocytosis, we propose that AMPARs are trafficked in REs outside the synaptic spine region (*Figure 5C*). Consistent with this trafficking role are studies demonstrating that different kinds of endosomes, including REs, travel long distances to new locations during recycling in dendrites and axons (*Yap and Winckler, 2012*). Furthermore, an EM study using three-dimensional reconstruction analysis found that in dendrites of rat, hippocampal neurons independent REs were not maintained at each spine. Instead, up to 20 spines shared a single recycling endosome (*Cooney et al., 2002*). In our model (*Figure 5C*), AMPAR exocytosis sites for the dynamin/clathrin-dependent pathway during cLTP are placed within the spine to deliver the AMPARs within the diffusible pool near the PSD of the synapse. In short, we propose that the role for the dynamin/clathrin-dependent recycling pathway during cLTP is to move AMPARs from sites distant from synaptic activation to sites near the activated synapse.

During cLTD, we propose that AMPAR recycling via the dynamin/clathrin-dependent recycling pathway is also increased similar to what occurs during cLTP. Recycling AMPARs in TfR-containing REs are trafficked in and out of synapses and along dendrites except in the opposite direction (*Figure 5D*).

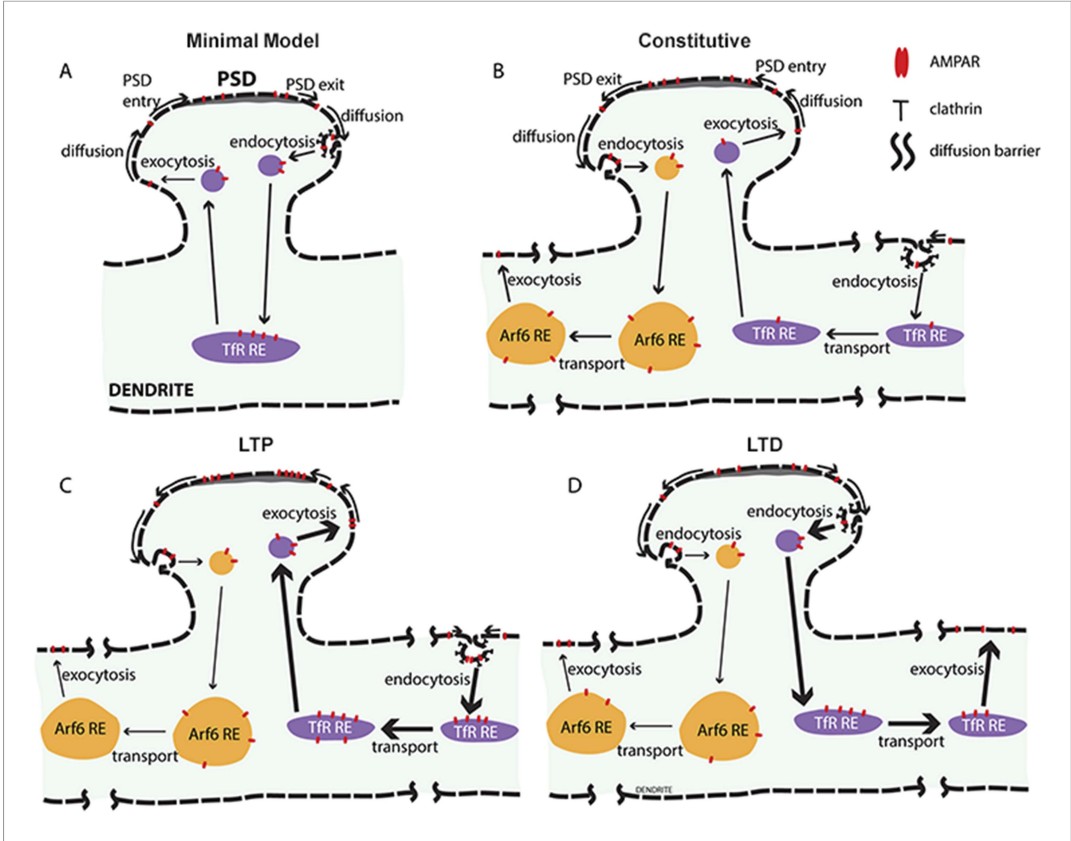

**Figure 5**. New model for AMPAR constitutive and activity-dependent recycling. (**A**) Conventional 'single-synapse' AMPAR recycling model. AMPARs exit post-synaptic densities (PSDs) by lateral diffusion and are endocytosed into clathrin-coated pits at sites near PSDs. After, endocytosed AMPARs are sorted into recycling endosomes (REs), the same pathway used by TfR for recycling back to the cell surface, where AMPARs diffuse to and could be trapped in PSDs. An underlying assumption of the model is that AMPAR recycling is restricted to single spines and the endosomal membranes in the spine and dendrite neighboring the spine. Activity during LTP increases AMPAR exocytosis and transport from REs without altering its endocytosis thereby decreasing AMPAR levels in REs. (**B**) AMPAR recycling model with two AMPAR-recycling pathways under constitutive conditions. Based on evidence in this study, at least two different AMPAR-recycling pathways exist for AMPAR recycling. AMPARs largely recycle through the Arf6- and TC10-dependent recycling pathway, which originates at sites near the PSD and endocytose at clathrin- and dynamin-independent sites. This recycling pathway acts to move AMPARs from sites near the PSD to sites distant from the synapse such that AMPARs cannot return via membrane diffusion. Smaller numbers of AMPARs endocytose at sites distant from the spine via clathrin-coated pits using dynamin. Endocytosed AMPARs traffic into REs in the same pathway as TfRs. This recycling pathway acts to move AMPARs from distant sites to sites accessible to PSDs so that AMPARs can diffuse into PSDs to balance the loss via the Arf6-dependent recycling pathway. (**C**) AMPAR recycling after cLTP. Activity-dependent events during cLTP increase AMPAR recycling through the dynamin-dependent pathway, trafficking AMPARs from clathrin-coated pits distant from the stimulated synapse to sites accessible to the stimulated PSD. Endocytosed AMPARs are transported in the REs with the net effect of trafficking more AMPARs into the stimulated PSD. AMPAR recycling through the Arf6- and TC10-dependent recycling pathway continues unchanged after cLTP. (**D**) AMPAR recycling after cLTD. Similar to what is observed after cLTP, activity-dependent events during cLTP increase AMPAR recycling through the dynamin-dependent pathway except with the net effect of trafficking AMPARs out of PSDs and away from the stimulated spines. AMPAR endocytosis occurs at clathrin-coated pits near stimulated PSD and AMPARs transported in REs away from the cLTD spines and recycling to the cell surface at distant sites. The Arf6- and TC10-dependent recycling pathway, shown in (**B**), is also unchanged during cLTD, and overall traffic AMPARs out of synaptic spines.

This is consistent with many studies that have found that AMPAR endocytosis increases via the dynamin/clathrin-dependent pathway. However, it has been assumed that during LTD, increased AMPAR endocytosis occurs without increased AMPAR exocytosis resulting in larger local endosomal stores of AMPARs. Instead, we suggest that during cLTD, AMPARs are transported away from

synapses. In support of this idea, Tao-Cheng et al., (2011) reported the appearance of AMPARs in clathrin-coated pits within synaptic spines, which was not observed during constitutive conditions or cLTP. This finding suggests that AMPARs near synapses are only removed locally via clathrin-coated pits during cLTD. In our model, AMPARs are trafficked during cLTD from clathrin-coated pits at activated spines via REs to distant sites. Overall, AMPARs flow out of synaptic spines during cLTD and into synaptic spines during cLTP. The function of the dynamin/clathrin-dependent recycling pathway is, thus, to traffic AMPARs from regions of low activity to regions of high activity and in this way the pathway underlies a Hebbian redistribution of AMPARs.

Our finding that TC10WT expression had opposite effects on synaptic AMPAR currents, a measure of functional synaptic AMPARs, and the cell-surface levels of dendritic AMPARs suggests that the Arf6/TC10-dependent recycling pathway traffics AMPARs out of synaptic spines. In this study, we conclude that TC10 regulates AMPAR recycling through the Arf6/TC10-dependent pathway. The simplest explanation for why TC10WT expression acts to increase cell-surface AMPARs levels is that it increases exocytosis from Arf6-dependent pathway REs so that fewer AMPARs are in the Arf6 endosomes and more AMPARs are in the plasma membrane. The decrease in the levels of synaptic AMPARs with TC10WT expression can be explained if the Arf6-dependent recycling pathway acts to traffic AMPARs via endocytosis and exocytosis from synapses to distant plasma membrane sites. For this reason, in our model, we have placed the endocytosis site of the Arf6-dependent recycling pathway near the PSD and the exocytosis site outside of the spine (Figure 5B). During constitutive AMPAR recycling, if AMPAR recycling via the Arf6-dependent recycling pathway acts to move synaptic AMPARs out of the spine and away from the synapse, then counteracting processes must be bringing other AMPARs back to the synapse in order to keep the levels of AMPAR constant. During cLTD (Figure 5D), we suggest that the Arf6/TC10-dependent recycling pathway acts together with the dynamin/clathrin-dependent pathway to traffic AMPARs away from synaptic spines. If the Arf6/TC10-dependent recycling pathway traffics AMPARs out of the spines then blocking it would act to reduce LTD, as we observed with TC10DN expression (Figure 4A–C). As explained above, the differential effects of TC10DN and TC10CA during LTD may arise from the differences in how they act during the Arf6/TC10-dependent recycling pathway. During constitutive conditions (Figure 5B) and during cLTP (Figure 5C), the Arf6/TC10-dependent recycling pathway would act counter to the dynamin/clathrin-dependent pathway, which in our model would traffic AMPARs into synaptic spines under these conditions. This aspect of our model explains why blocking Arf6/TC10-dependent recycling pathway with the TC10DN expression may actually increase levels of LTP (Figure 4A–C).

## Materials and methods

### Antibodies

The following primary antibodies were used: anti-RFP (MBL, rabbit, #PM005), anti-GM130 (BD Biosciences, mouse, 1:500), anti-transferrin receptor (Zymed, Thermo Scientific Pierce, Waltham, MA, #13-6800, 1:300), anti-synapsin (Chemicon, EMD Millipore, Temecula, CA, mouse, 1:500), anti-Bassoon (Synaptic Systems, Goettingen, Germany, Guinea Pig, 1:300), anti-GFP (Sigma-Aldrich, St. Louis, MO, rabbit, 1:5000), anti-GluR1 (Millipore, EMD Millipore, Billerica, mouse; CalBioChem, Merck Millipore, Temecula, CA, rabbit). Dynasore was from Sigma (D7693); Sulfo-NHS-SS-biotin was from Pierce.

### cDNA cloning and mutagenesis

Human TC10 constructs were obtained from Dr. J. Marshall (Brown University), and then subcloned into the pSP2 vector, containing a CMV promoter and Venus tag (a brighter and more photostable YFP variant) to generate fusion proteins. mCherry-GluA1 construct was obtained from Dr. C. Garner (Stanford University). Rat TC10, Cdc42, and Rab11 genes were cloned from mRNA of 20 DIV cortical neuron cultures by RT-PCR with the following primers: TC10: FWD (with EcoRI) 5′-CCTTACATAGAATTCATGGCTCACGGGCCC-3′, REV (with BamHI) 5′-GGCCCAGTGGATCCTCACGTAATCAAACAACAGTTTATAC-3′; Cdc42: FWD (with EcoRI) 5′-CGTTACTAAGAATTCATGGGCACCCGCGAC-3′, REV (with BamHI) 5′-GGCCTCGACGGATCCTTAGATGTTCTGACAGCACTGC-3′; Rab11: FWD (with EcoRI) 5′-CGTTACTAAGAATTCATGGGCACCCGCGAC-3′, REV (with BamHI) 5′-GGCCTCGACGGATCCTAAGATGTTCTGACAGCACTGC-3′. The genes were then inserted into a customized vector (originated from pEYFP from Clontech, Takara, Mountain View, CA) with a CMV

promoter and mCherry tag. The lentiviral vector, FUGW, and the helper plasmids, Δ8.9 and VSVg, were obtained from Dr. C. Garner (Stanford University) and were used to clone all genes listed above for production of lentiviruses. The 3 candidate RNAi constructs for TC10 were purchased from Sigma–Aldrich.

## Primary neuronal culture and transfections

Rat E18 hippocampal neurons were cultured on poly-L-lysine treated coverslips in Neurobasal medium supplemented with NS21 and GlutaMAX. At 14–17 DIV, neurons were transfected by Lipofectamine 2000 with serum-free Neurobasal medium. The amount of cDNA transfected ranges from 1 to 2 µg per coverslip (d = 12 mm) as needed. Hippocampal cultures were prepared using Neurobasal medium, 2% (vol/vol) B27, and GlutaMAX. Briefly, hippocampi from embryonic (E18–19) Sprague Dawley rats of either sex were dissected, dissociated in 0.05% trypsin (vol/vol, Life Technologies, Thermo Fisher Scientific, Grand Island, NY), and cells were plated at a density of $\sim 4 \times 10^5$ cells/mL on poly-L-lysine-coated 12-mm coverslips. Coverslips were maintained in Neurobasal medium containing B27 and GlutaMAX (all from Life Technologies).

Neuronal cultures were transfected at 14–17 DIV with the Lipofectamine 2000 transfection reagent (Life Technologies) according to manufacturer's recommendations, with the exception that 1–2.5 µg of each cDNA in 62.5 µl of Neurobasal media and 2.0 µl of Lipofectamine 2000 in 62.5 µl of Neurobasal media were mixed and added to coverslips in 12-well plates.

## Production of lentiviruses and infection of neurons

Freshly thawed HEK cells were cultured in DMEM + 10% fetal bovine serum (FBS) medium. Lentiviral plasmid and the helper plasmids were transfected using $Ca_3(PO_4)_2$ method. $\sim 50$ hr after transfection, supernatant containing the virus particles was collected. After a brief centrifugation to remove cell debris, the virus solution was then mixed with PEG 8000 to incubate at 4°C overnight, followed by centrifugation at 4000 rpm for 30 min at 4°C. The virus pellet was then resuspended in cold phosphate buffered saline (PBS). At 0–1 DIV, neurons were infected with high-titer lentiviruses, and 2 days post-infection, culture medium was changed with fresh Neurobasal, B27, and GlutaMAX.

## Immunofluorescence staining and microscope imaging

For surface labeling, primary antibody was added into the culture medium and incubated at room temperature for 30 min, or for 20 min at 37°C. Cells were then washed with PBS and fixed with 4% PFA/4% sucrose/PBS for 10 min, then incubated with blocking solution (2% glycine, 1% BSA, 0.2% gelatin, 0.5M $NH_4Cl$, PBS) for 1 hr and secondary antibody in blocking solution at room temperature for 1 hr. For internalized staining (antibody-feeding assay), primary antibody was added into the culture medium and incubated at 37°C for 40 min, then cells were washed with acid wash buffer (0.5M NaCl, 0.5% acetic acid, pH2) for 30 s, and fixed. Cells were then incubated with blocking solution and secondary antibody. For permeabilized staining, after fixation, the cells were permeablized in 0.1% Triton/PBS for 5–10 min, blocked and stained with primary and secondary antibodies. The stained coverslips were mounted to glass slides with ProLong Gold (Life Technologies) mounting media and left in dark to harden overnight.

For internalized staining of Tf-Alexa 647, Tf-Alexa 647 was added into the culture medium and incubated at 37°C for 1 hr. Then, cells were washed with acid wash buffer and fixed for imaging.

All images were taken on either an Olympus DSU or Marianas Yokogawa type spinning disk confocal microscope with back-thinned EMCCD camera. Z-stack slices were taken with 0.2-µm step size, with 5–10 slices for each cell. For the surface staining experiments, z-plane limits for acquisition were determined by the surface staining on dendrites; for the assay to measure changes in Golgi/endosomal/synaptic localization of GluA1, the z-plane limits were set according to organelle marker staining (GM130, transferrin receptor or Arf6, and synapsin, respectively).

## cLTP and cLTD protocols

To induce chemical LTP, DIV17 neurons were washed in $Mg^{2+}$ free buffer (in mM: NaCl 150, $CaCl_2$ 2, KCl 5, HEPES 10, glucose 30, strychnine 0.001, bicuculline 0.02) 3 times, and incubated in glycine buffer ($Mg^{2+}$ free buffer with 0.2 mM glycine) at 37°C for 15 min. Then, $Mg^{2+}$ buffer ($Mg^{2+}$ free buffer with 2 mM $MgCl_2$) was added to block NMDARs and cells were incubated at 37°C for 30 min before live surface labeling with anti-RFP.

To induce chemical LTD, 14 DIV neurons were washed in $Mg^{2+}$ free buffer 3 times, and incubated in NMDA buffer ($Mg^{2+}$ free buffer with 0.02 mM NMDA) at 37°C for 5 min. Then, $Mg^{2+}$ buffer was added and cells were incubated at 37°C for 1 hr before live surface labeling.

## Quantification of images and statistical tests

Image quantification was performed using NIH ImageJ software. To calculate the surface/total ratio for exogenously expressed GluA1, all images of the same channel were first background subtracted using the same averaged value, which was measured manually across images (with variation <0.5%). The sum of pixel intensity for the z-stack was calculated using the 'sum of slices' and 'histogram' functions, excluding zero-intensity pixels. The surface/total ratio was then calculated as the ratio of the intensity of surface channel to total channel.

Surface expression of endogenous GluA1 was quantified using 'sum of slice' z-projections of images. Each field was background subtracted, and mean intensities were normalized to YFP control.

To measure the Golgi localization of GluA1, the images were background subtracted with the same method described above, and then, the GM130 channel image was thresholded and transformed into a binary mask used to measure the pixel intensity of GluA1 in each slice of the stack. The average intensity of the processed z-stack image was then measured by selecting the cell of interest manually (if more than one cell was present in the image) and applying the 'measure' function.

To measure the degree of co-localization of GluA1 with TfR-positive endosomes, images were assigned a random number and analyzed blindly. Analysis of the co-localization of endogenous GluA1 with TfR was carried out by background subtracting and thresholding image fields so that only puncta that were twofold greater than background were selected. Co-localizing puncta were evaluated using the Analyze Particles function in ImageJ.

To measure the degree of overlap between endogenous GluA1 and Arf6 sub-compartments, we compared fluorescence signals above background in both channels along manually outlined segments of dendrites. Pearson's correlation coefficients were generated for background-subtracted image pairs using the Intensity Correlation Analysis plugin in ImageJ. A similar approach was used to measure the degree of overlap between TC10 and Arf6. Thresholded GluA1 punctal size and density in dendritic shafts was analyzed using the Analyze Particle function in ImageJ.

Statistical comparisons were made using two-tailed Student's $t$ tests or ANOVA/Tukey *post hoc* analysis as indicated. Statistical graphs were generated with Graphpad Prism or StatPlus software.

## Whole-cell recording

Dual whole-cell recordings were performed at DIV12–15 on primary dissociated hippocampal cultures transfected with Venus-TC10DN or Venus-TC10CA. Neurons were bathed in carbogen (95% $O_2$, 5% $CO_2$) bubbled ACSF (in mM: 120 NaCl, 3 KCl, 2 $CaCl_2$, 1.25 $NaH_2PO_4$, 2 $MgSO_4$, 20 D-(+)-glucose, 26 $NaHCO_3$). The internal solution consisted of (in mM): 120 K gluconate or Cs gluconate, 40 HEPES, 5 $MgCl_2$, 2 NaATP, 0.3 NaGTP. Recordings were performed at room temperature (21°C). Hippocampal cultures were mounted on an Olympus microscope (BX51WI) and visualized using differential interference microscopy. Transfected neurons were visualized via excitation at 530–550 nm. Electrode resistance was between 5 and 10 MΩ. Patch-clamp recordings were obtained using a MultiClamp 700B Commander (Molecular Devices, Sunnyvale, CA). Data acquisition and analysis were performed using AxoGraph X (AxoGraph Scientific, Sydney, Australia) and pCLAMP 9 (Molecular Devices) software. Events were sampled at 10 kHz and low-pass filtered at 2 kHz. Series resistance (Rs) was monitored throughout all experiments, and results were not included if significant variation (>20%) occurred during any experiment. Action potentials were induced in presynaptic neurons by a 20-ms current injection of 20–100 pA. AMPAR EPSCs in response to presynaptic action potentials were collected at 0.1 or 0.2 Hz. Statistical significance of changes in AMPAR EPSC amplitudes was tested using Student's t test with a level of significance set at $p < 0.05$.

## Biotinylation and isolation of the internalized pool

Cortical neurons were pretreated with 80 µM dynasore or 1% DMSO for 40 min at 37°C. Cells were then washed, and incubated with 0.5 mg/mL Sulfo-NHS-SS-Biotin at 4°C for 30 min, and excessive Sulfo-NHS-SS-Biotin was washed off. Cells were then incubated at 37°C for 1 hr in the presence of 80 µM dynasore or 1% DMSO. Then Sulfo-NHS-SS-Biotin on the cell surface was cleaved with glutathione. Cells were then harvested and lysed; biotinylated proteins were pull down using streptavidin–sepharose beads and were analyzed on Western blot.

## Acknowledgements

We are grateful to Dong Li for performing supporting experiments. We thank Ning Zheng's thesis committee members (Drs. Benjamin Glick, Richard Fehon, and David Kovar) for instructive guidance and support during the completion of the dissertation project. We thank Vytas Bindokas for his technical support with the analysis of fluorescent microscopy experiments. This work was supported by the U.S. National Institutes of Health under grant numbers NS043782, NS090903 and DA035430 (WNG). This project was also supported by the University of Chicago, Department of Neurobiology Erma Smith Scholarship from the Physiology Endowment Fund and faculty fellowships to WNG and JMM from the Marine Biological Laboratory. The funders had no role in study design, data collection and analysis, decision to publish, or preparation of the manuscript.

## Additional information

### Funding

| Funder | Grant reference | Author |
|---|---|---|
| National Institutes of Health (NIH) | NS043782 | William N Green |
| Marine Biological Laboratory (MBL) | Faculty fellowships | Johanna M Montgomery, William N Green |
| University of Chicago (UChicago) | Erma Smith Scholarship | Johanna M Montgomery, William N Green |
| National Institutes of Health (NIH) | DA035430 | William N Green |
| National Institutes of Health (NIH) | NS090903 | William N Green |

The funders had no role in study design, data collection and interpretation, or the decision to submit the work for publication.

### Author contributions

NZ, Conception and design, Acquisition of data, Analysis and interpretation of data, Drafting or revising the article; OJ, Acquisition of data, Analysis and interpretation of data, Drafting or revising the article; CM, Acquisition of data, Analysis and interpretation of data; JMM, WNG, Conception and design, Analysis and interpretation of data, Drafting or revising the article

### Ethics

Animal experimentation: We followed AVMA guidelines to prevent pain and suffering of animals, and only used the minimum number of animals necessary to obtain conclusions in our experiments. Any pain, discomfort, or distress associated with the surgical procedures was prevented by the administration of the volatile anesthetic, Isoflurane. All animal procedures have been approved by the University of Chicago Institutional Animal Care and Use Committee (IACUC/ACUP protocol #72016) and are in accordance with the recommendations of the Panel on Euthanasia of the American Veterinary Medical Association.

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
