## [Decision Letter]

Thank you for sending your work entitled “Synaptic Activity Regulates AMPA Receptor Trafficking Through Different Recycling Pathways” for consideration at *eLife*. Your article has been favorably evaluated by Vivek Malhotra (Senior editor), a Reviewing editor, and two reviewers.

The Reviewing editor in consultation with other reviewers has assembled the following comments to help you prepare a revised submission.

1) All the figures with micrographs are very small on the page and hard to visualize. This may be partially technical (figure quality) and thus the figures need improvement.

2) The ability of TC10 to alter constitutive AMPAR trafficking, the lack of overlap of the constitutive AMPAR pathway and the transferrin pathway and the selective effect of dynasore on the transferrin pathway appear to be convincing. However, it gets a little muddy when TC10 and LTP do not overlap, but TC10 partially overlaps LTD. This needs an explanation.

3) Moreover, the experimental evidence for these two pathways is primarily based on overexpressed and tagged proteins. Although not an absolute requirement, the authors should at least try to show that endogenous GluA1 obeys the trafficking rules with TC10 and its mutants that they report for exogenous GluA1.

4) According to the model, one would expect a greater proportion of AMPARs colocalizing with the TC10/Arf6 machinery before LTP; LTP induction should shift this towards the clathrin pathway. The authors therefore need to show AMPAR distribution prior to and post LTP induction using antibody staining for the two endocytic pathways, if possible. There are good ABs for the clathrin pathway (which have been used in the literature), the Arf6/TC10 ABs are. They are available from Abcam, for example.

5) It should be demonstrated that the chemLTP protocol indeed works, i.e. potentiates EPSC amplitude and that endogenous AMPARs shift after successful LTP induction from TC10/Arf6 to clathrin.

[Editors' note: a previous version of this study was rejected after peer review, but the authors submitted for reconsideration. The previous decision letter after peer review is shown below.]

Thank you for choosing to send your work entitled “Synaptic Activity Regulates AMPA Receptor Trafficking Through Different Recycling Pathways” for consideration at *eLife*. Your full submission has been evaluated by Vivek Malhotra (Senior editor) and four peer reviewers, one of whom is a member of our Board of Reviewing Editors, and the decision was reached after discussions between the reviewers. We regret to inform you that your work will not be considered further for publication.

Although there was some difference in the reviewers' assessment of the relevance of your study, they agreed that the problem addressed is definitively of interest. However, the experimental work that is required in the reviewers' opinion is too extensive to allow for a revision according to *eLife* policy. As you can see from the comments below, the reviewers found several sets of data inconsistent with the conclusions. Furthermore, most of them agree that the conclusions go beyond what is supported by the data, and technical issues were also raised. Please note, however, that this feedback is not meant to discourage you from pursuing this line of research further, and they would be interested in looking at the work again if the experiments are further developed.

*Reviewer #1*:

1) An essential part of the study uses colocalization (at the LM level) of GFP-tagged and overexpressed AMPA-receptors with organelle markers, most prominently with Arf6 and transferrin-receptor, as read-out for identifying the subcellular localization. However, the way colocalization is analyzed is highly problematic. This is exemplified in Figure 3: Here Arf6 yields a granular but rather diffuse staining whereas GluA1 is punctate. Considering the diffuse staining of Arf6, a high degree of colocalization would be scored anywhere along the dendrite. Obviously, this problem is owing to the limited resolution (as no super-resolution techniques were employed). At least, the authors should analyze the images by cross-correlation analysis using line-scans along the dendrite. Another (although related) method may be to move all Glu-A1 spots by 500 nm along the central axis of the dendrite and then re-evaluate colocalization. This degree of colocalization can then be used as correction factor (random vs. specific colocalization). Considering the images provided, I am not at all convinced that GluA1 and Arf6 indeed reside on the same organelle, at least not to the high degree claimed in the manuscript. By the way, what is the value of colocalization if the analysis is done inversely (percentage of Arf colocalizing with GluA1)?

This problem affects all colocalization analysis and thus constitutes a serious problem of the study. For Arf6 and TC10 the colocalization appears to be more conspicuous but again this needs to be quantified by cross-correlation analysis (using line-scans).

2) Overexpression of multi-subunit and multi-spanning membrane proteins in any cell type is problematic since the folding and trafficking pathways are easily overloaded. Indeed, the pattern of mCherry-GluA1 gives rise to concerns: The accumulation in what appears to be the ER hints towards a “jam” in trafficking that may be related to folding problems or to an overload of the trafficking system. Under such overload conditions, proteins are known to “spill-over” towards other organelles, and they may cause changes in the ratio between surface and endogenous pools. As control, stainings should be carried out for the endogenous receptor (both using untransfected neurons and neurons transfected with the reporter construct) to ensure that there is no perturbation of trafficking due to the overexpression.

3) Another problem (at least in my view) is that in the stainings synapses cannot be unequivocally identified (which could be done e.g. by colocalization with synaptic markers). Thus, the conclusions suggesting that the two recycling pathways differ in their subcellular localization (clathrin-dependent and activity-dependent: synaptic, constitutive: extrasynaptic) is not directly proven.

4) The authors state that dynasore treatment does not affect GluA1 endocytosis, in contrast to the endocytosis of transferrin. These data appear to be convincing. However, when comparing sham and dynasore treated neurites in Figure 4 dynasore appears to cause substantial change in the GluA1 staining pattern (much more diffuse). What is the reason for this? Does the Arf6-staining pattern (which, according to the authors, should label the same compartment) change in a comparable manner?

*Reviewer #2*:

First, they show that knock-down of TC10 reduces surface expression of exogenous mCherry-GluA1. Similarly, expressing dominant negative DN and const. active CA TC10 reduces surface GluA1. The authors should offer an explanation as to why both mutants have produced the same phenotype. Also, in addition to monitoring fluorescent exogenous GluA1 it would be worth determining whether the effect of TC10 KO and TC10 mutant-expression holds for endogenous AMPARs as one would expect (using a surface biotinylation protocol, for example).

The result of the paired recordings is somewhat puzzling. If, as they suggest, TC10 indeed differentiates between surface vs synaptic AMPAR it would be worth considering to determine AMPAR levels in somatic patches (which are expected to be elevated in response to TC10 WT expression).

*Reviewer #3*:

There are two sorts of data in this paper. One is the effect of a TC10 mutant on the level of GluA1 in a cell compartment or on the extent of a process, such as cLTP. Specifically, the authors express wild type TC10 (WT) and dominant negative (DN) and constitutively active (CA) TC10 mutants and carry out knockdown (KD) of TC10 to dissect GluA1 trafficking. It is confusing that DN and CA expression and KD can have the same effects, decreasing dendritic GluA1 surface levels, while WT has the opposite effect, increasing dendritic GluA1 surface levels (Results section and Figure 1). Why should enhancing and blocking TC10 function have the same result? The authors state that because the DN and CA mutants reduced surface expression to the same extent, the AMPARs lost from the Arf6 endosomes must be redistributed to an unidentified intracellular compartment. Isn't it possible that overexpression interferes with trafficking non-specifically?

The other is the effect of a mutant on the colocalization of GluA1 with a marker, in particular with transferrin endocytosed by the transferrin receptor. This latter form of data seems more compelling to me and on the basis of such data they authors argue that TC10 functions in GluA1 trafficking associated with Arf6 rather than with the transferrrin receptor, which is a marker for clathrin/dynamin mediate recycling/secretory pathway. The authors conclude that two pathways operate.

Making a case for the operation for the two pathways is important. And for the most part, the data seem respectable. However, the logic for interpretations of the data often escape me, as is the case with similar effects of DN, CA expression and KD of TC10. Also, the authors completely neglect a vast amount of literature that states that GluA1 trafficking is controlled by phosphorylation of the C-terminal domain of the subunit. The authors suggest that TC10 functions in controlling the interaction of nPIST with TARPS that are in complexes with AMPA receptors. But there is abundant evidence that GluA1 phosphorylation, such as takes place during LTP, regulates GluA1 trafficking to and from the surface. The literature does not distinguish which pathway is affected by this phosphorylation (clathrin dependent versus independent), and in this sense the paper contributes a new view. Yet it does so while overlooking the fact that phosphorylation controls the trafficking, be it TC10-dependent or independent.

*Reviewer #4*:

Figure 1. Pictures of neurons only show soma (Figure 1) and it would be useful to show the entire neuron, at least in the first figure. Notably, this is all overexpression data and the reader would be enlightened if they also examined endogenous AMPA-R levels. (They do look at endogenous GluA1 later on in terms of localization with known recycling markers). Another concern: discrepancy between electrophysiology and surface labeling of AMPA-Rs for TC10WT vs mutants.

Figure 2. Based on title of this figure, the actual supporting data are limited. For instance denditric shaft primary data are only shown for TC10DN and not other conditions, but phenomenon of dendritic accumulation is generalized to all. This is also the only positive data in the figure (other panels are negative) so the authors should be more thorough. The GM130 colocalization data take up most of figure, and doesn't add a lot. Overall it isn't a very thorough investigation of potential effects on forward trafficking.

Figure 3 and Figure 4. One concern: DN and CA have different effects on colocalization yet have same effects on AMPA-R accumulation in dendrites?

Also, the biochemistry in Figure 4 is poorly controlled. The authors should show total protein amounts and also negative control (intracellular protein).

Figure 5. The TC 10DN mutant seems to affect LTD, but not LTP- however the authors follow up on LTP showing that it involves dynamin dependent pathways, while neglecting to follow up on the LTD results. The LTD effect I mostly ignored when summarizing their results overall. Also, it is essential for the authors to include representative images along with any quantifications.

Figure 6. It is nice to have a model figure, but this one is too far reaching. A simpler model included in another figure might be more appropriate.

---

## [Author Response]

*1) All the figures with micrographs are very small on the page and hard to visualize. This may be partially technical (figure quality) and thus the figures need improvement*.

We have improved all of the figures as requested in the new submission.

2) The ability of TC10 to alter constitutive AMPAR trafficking, the lack of overlap of the constitutive AMPAR pathway and the transferrin pathway and the selective effect of dynasore on the transferrin pathway appear to be convincing. However, it gets a little muddy when TC10 and LTP do not overlap, but TC10 partially overlaps LTD. This needs an explanation.

We have provided a more detailed explanation in the Discussion section.

3) Moreover, the experimental evidence for these two pathways is primarily based on overexpressed and tagged proteins. Although not an absolute requirement, the authors should at least try to show that endogenous GluA1 obeys the trafficking rules with TC10 and its mutants that they report for exogenous GluA1.

We performed the experiment requested by the reviewer in the previous resubmission (see new Figure 1). Because our primary interest was the recycling of AMPARs on dendrites, we tested the effects of TC10WT, TC10DN and TC10CA on the endogenous AMPARs on dendrites using an antibody that recognizes an extracellular epitope on GluA1. We obtained results basically identical to the results obtained with the transfected Cherry-GluA1 (compare Figure 1).

4) According to the model, one would expect a greater proportion of AMPARs colocalizing with the TC10/Arf6 machinery before LTP; LTP induction should shift this towards the clathrin pathway. The authors therefore need to show AMPAR distribution prior to and post LTP induction using antibody staining for the two endocytic pathways, if possible. There are good ABs for the clathrin pathway (which have been used in the literature), the Arf6/TC10 ABs are. They are available from Abcam, for example.

In Figure 4—figure supplement 1, we display images of neurons after cLTP or sham-treated. The neurons are stained for internalized mCherry-GluR1 and either native transferrin receptors or for Arf6-HA.

We have attempted, without success, to stain the neurons with several Arf6/TC10 antibodies. We have discussed our lack of results with Dr. Julie Donaldson (NIH), who is an expert in this area. She also has had significant difficulty staining neurons with Arf6/TC10 antibodies.

5) It should be demonstrated that the chemLTP protocol indeed works, i.e. potentiates EPSC amplitude and that endogenous AMPARs shift after successful LTP induction from TC10/Arf6 to clathrin.

This experiment has already been performed by Dr. Jo Montgomery, who is the electrophysiologist on the manuscript. In a previous paper (Li D, Specht CG, Waites CL, Butler-Munro C, Leal-Ortiz S, Foote JW, Genoux D, Garner CC, Montgomery JM (2011) SAP97 directs NMDA receptor spine targeting and synaptic plasticity. J Physiol 589:4491– 4510) in Figure 1, Dr. Montgomery showed using the same chemLTP protocol as in our manuscript, and a similar hippocampal neuronal culture preparation, that the chemLTP protocol works with respect to potentiating EPSC amplitude long-term. We have added a sentence to the manuscript pointing this out (please see the subsection headed “Changes in synaptic activity alter the recycling pathway taken by AMPARs”).

[Editors’ note: the author responses to the previous round of peer review follow.]

The manuscript had previously been rejected with the provision that you “would be interested in looking at the work again if the experiments are further developed”. We believe that the experiments have been further developed with several new experiments and significant new analysis of the data performed and added to the manuscript. We feel that these additions address all of the reviewers' concerns, and also provide added insights into the details of how TC10 acts to regulate the recycling of AMPA receptors at synapses, a major concern of the reviewers.

We also feel that there is some misunderstanding of the significance of our findings. How AMPA receptors are recycled at synapses is an important issue in neuroscience because it is one of the main mechanisms by which long-lasting synaptic changes occur with changes in activity. The consensus in the field is that AMPA receptor recycling occurs through the following single pathway (see Figure 4). First, endocytosis occurs through the clathrin-dependent pathway taken by transferrin receptors. Second, the endocytosed AMPA receptors, like transferrin receptors, remain in recycling endosomes prior to their exocytosis. Third, exocytosis from recycling endosomes returns AMPA receptors to the same synapse from which endocytosis occurs. This model predicts that increases in AMPA receptors at synapses occur by increased exocytosis from recycling endosomes. Decreases in AMPA receptors at synapses occur by increased AMPA receptor endocytosis from the synapse. The model is so well accepted that a number of labs use fluorescently tagged transferrin receptors, which are much more abundant at synapses than AMPA receptors, as a substitute for AMPA receptors in their trafficking studies. Based on our results, transferrin receptor trafficking is quite different from that of AMPA receptors.

We believe that the results reported in our manuscript are significant because they challenge this conventional model of AMPAR recycling and provide new information about how AMPA receptors recycle at synapses. We have significant evidence that AMPARs recycle through at least two different pathways. One pathway is routed through endosomes lacking transferrin receptors but containing Arf6 and TC10. This pathway predominates without synaptic activity paradigms that cause long-term potentiation (LTP) and long-term depression (LTD). Also, trafficking via this pathway is reduced by the TC10 mutants with the GDP-bound mutation, which causes AMPA receptors to pile up in the endosomes, and the GTP-bound mutation, which causes AMPA receptors to exit the endosomes but pile up in transport vesicles prior to fusion with the plasma membrane. LTP and LTD do not appear to affect recycling through this pathway but do affect recycling through the second pathway, which is the same recycling pathway taken by transferrin receptors. Contrary to the currently accepted model, increased AMPAR levels caused by LTP caused large increases in AMPA receptor endocytosis. These data indicate that the second recycling pathway is regulated by synaptic activity. It also suggests that AMPA receptors are trafficked into the synapse via the second recycling pathway from regions outside the synapse.

Reviewer #1:

*1) An essential part of the study uses colocalization (at the LM level) of GFP-tagged and overexpressed AMPA-receptors with organelle markers, most prominently with Arf6 and transferrin-receptor, as read-out for identifying the subcellular localization. However, the way colocalization is analyzed is highly problematic. This is exemplified in*
Figure 3: *Here Arf6 yields a granular but rather diffuse staining whereas GluA1 is punctate. Considering the diffuse staining of Arf6, a high degree of colocalization would be scored anywhere along the dendrite. Obviously, this problem is owing to the limited resolution (as no super-resolution techniques were employed). At least, the authors should analyze the images by cross-correlation analysis using line-scans along the dendrite.*

This degree of colocalization can then be used as correction factor (random vs. specific colocalization). Considering the images provided, I am not at all convinced that GluA1 and Arf6 indeed reside on the same organelle, at least not to the high degree claimed in the manuscript. By the way, what is the value of colocalization if the analysis is done inversely (percentage of Arf colocalizing with GluA1)?

This problem affects all colocalization analysis and thus constitutes a serious problem of the study. For Arf6 and TC10 the colocalization appears to be more conspicuous but again this needs to be quantified by cross-correlation analysis (using line-scans).

To address the reviewer's concerns we repeated the experiments (in former Figure 3, now new Figure 2) and were able to obtain significantly better resolution of the Arf6 immunofluorescence, as well as performing the analysis along the lines suggested by the reviewer. The Arf6 fluorescence was better resolved using a spinning disk confocal outfitted with a 100x Alpha Plan-Fluar/NA 1.45 objective. Instead of the diffuse staining, Arf6 in the new images is clearly found in an intracellular sub-compartment in dendrites from which co-localization with other proteins was better measured. The TC10 and GluA1 fluorescence signals were also better resolved. For GluA1, the improved resolution revealed clear differences in the size and number of GluA1 puncta, as now analyzed in new Figure 2.

As suggested by the reviewer, we re-analyzed the data in the figure using cross-correlation analysis in order to better assess co-localization of GluA1 with Arf6 and different TC10 constructs. The figures relevant to this concern are all in a separate rebuttal figure (see below). We measured the extent of overlap between channels using a specially developed algorithm that yields Pearson's correlation coefficients for dendrites in each image pair. This was achieved by comparing fluorescence signal above background in both channels, along manually outlined segments of dendrites. This is a standard method for evaluating the correlation of pixel intensity distributions between channels (Figure 6). We also performed a second, independent method of analysis to assess the degree to which GluA1 is enriched in Arf6-positive endosomes (Figure 6). We thresholded and masked dendritic Arf6 signal then measured GluA1 pixel intensities in these structures. Using this method, we measured a significant loss in GluA1 signal in Arf6 endosomes following TC10CA expression, also reflected by the loss of spatial overlap indicated by the decrease in Pearson's correlation coefficient. Both methods agreed that there were sizable increases and decreases in the distribution of GluA1 in Arf6 endosomes following expression of TC10DN and TC10CA mutants, respectively (Figure 2). Cross-correlation analysis demonstrated that these were statistically significant changes (Figure 2, Figure 6).

Author response image 1.Improved co-localization analysis. A. Quantification of the overlap between GluA1 and Arf6. Newly acquired images (examples in Figure 2) were analyzed to measure the Pearson's correlation coefficients (R_r_) of GluA1 co-localization with Arf6. Data are shown as mean ± SEM; TC10WT 0.46 ± 0.04; TC10DN 0.66 ± 0.04; TC10CA 0.31 ± 0.04 (n= 7-10 cells per group; *p<0.02 relative to TC10WT; **p<0.05 relative to TC10WT). B. A second method of analysis. GluA1 pixel intensity was measured in thresholded and masked Arf6 sub-compartments. Data are shown as mean ± SEM (normalized to TC10WT); TC10WT 1.0 ± 0.11; TC10DN 1.3 ± 0.10; TC10CA 0.56 ± 0.09 (n= 8-10 cells per group; **p<0.03 relative to TC10WT. C, Representative image of Arf6-HA and GluA1 in the dendritic shaft of a TC10DN expressing neuron (left panels), and with the GluA1 channel shifted along the central axis of the dendrite by ∼500nm (right panels). Bottom panels display line scans of pixel intensities for both channels along the yellow line in the dendrite. D, Change in Pearson's correlation coefficients following lateral shift of GluA1 channel in (C). Data are shown as mean ± SEM; TC10DN 0.66 ± 0.04; TC10DN shifted pixels 0.37 ± 0.05 (n=7 cells per group; *p<0.002).**DOI:**
http://dx.doi.org/10.7554/eLife.06878.018

Another (although related) method may be to move all Glu-A1 spots by 500 nm along the central axis of the dendrite and then re-evaluate colocalization.

We performed the analysis requested by the reviewer, shown in detail in Figure 6. We used the method suggested by the reviewer to determine the specificity of any measured co-localization. Expression of TC10DN yielded the highest co-localization between GluA1 and Arf6 (R_r_ = 0.7; Figure 2). We shifted the GluA1 channel along the central axis of dendrites by ∼500 nm and re-evaluated co-localization. Line scan analysis (Figure 6; bottom panels) illustrated an expected shift and mismatch in intensity peaks for the two channels along dendrites, and a corresponding and significant decrease in the Pearson's correlation coefficient (R_r_ = 0.37; Figure 6). These findings also support the significance of our methods used to measure specific co-localization in these dendritic sub-compartments.

2) Overexpression of multi-subunit and multi-spanning membrane proteins in any cell type is problematic since the folding and trafficking pathways are easily overloaded. Indeed, the pattern of mCherry-GluA1 gives rise to concerns: The accumulation in what appears to be the ER hints towards a “jam” in trafficking that may be related to folding problems or to an overload of the trafficking system. Under such overload conditions, proteins are known to “spill-over” towards other organelles, and they may cause changes in the ratio between surface and endogenous pools. As control, stainings should be carried out for the endogenous receptor (both using untransfected neurons and neurons transfected with the reporter construct) to ensure that there is no perturbation of trafficking due to the overexpression.

We performed the experiment requested by the reviewer. In Figure 1—figure supplement 4, representative images of somata are shown comparing staining for endogenous GluA1 subunits with that for the transfected Cherrry-GluA1. For both sets of neurons, the ER was simultaneously stained with an antibody against the ER marker, PDI. In both sets of neurons, the ER looks normal and GluA1 subunits are clearly largely found in the ER membrane. No evidence of a “jam” in trafficking was observed.

3) Another problem (at least in my view) is that in the stainings synapses cannot be unequivocally identified (which could be done e.g. by colocalization with synaptic markers). Thus, the conclusions suggesting that the two recycling pathways differ in their subcellular localization (clathrin-dependent and activity-dependent: synaptic, constitutive: extrasynaptic) is not directly proven.

This concern appears to be a misunderstanding of our proposed model of AMPAR recycling. We did not propose that the two recycling pathways differ in their subcellular localization (clathrin-dependent and activity-dependent: synaptic, constitutive: extrasynaptic). There were no experiments where we use staining to distinguish synaptic vs extrasynaptic locations, and we completely agree with the reviewer that the staining methods used in this manuscript are inappropriate to distinguish synaptic vs. extrasynaptic locations. What may have confused the reviewer is that we distinguished synaptic from total surface AMPARs (Figure 1) by comparing synaptic currents with cell-surface staining. Attempts have been made to make these points clearer in the manuscript.

4) The authors state that dynasore treatment does not affect GluA1 endocytosis, in contrast to the endocytosis of transferrin. These data appear to be convincing. However, when comparing sham and dynasore treated neurites in Figure 4 dynasore appears to cause substantial change in the GluA1 staining pattern (much more diffuse). What is the reason for this? Does the Arf6-staining pattern (which, according to the authors, should label the same compartment) change in a comparable manner?

We apologize for the image previously displayed in old Figure 4 (now new Figure 3) for the dynasore-treated neurons. We have repeated the experiments and carefully compared the GluA1 staining for sham and dynasore treated neurons using a spinning disk confocal outfitted with a 100x Alpha Plan-Fluar/NA 1.45 objective. We do not find significant differences between the two conditions as the previous image may have suggested. The previous image appears to have been somewhat out of focus. In the representative images in new Figure 3, dynasore treatment appears to have no effect on the distribution of GluA1, but TfR distribution appears to be altered, consistent with the dynasore-induced reduction in its internalization as shown in Figure 3 and Figure 3.

Reviewer #2:

First, they show that knock-down of TC10 reduces surface expression of exogenous mCherry-GluA1. Similarly, expressing dominant negative DN and const. active CA TC10 reduces surface GluA1. The authors should offer an explanation as to why both mutants have produced the same phenotype.

This is an excellent question that was also a concern of Reviewers 3 and 4. To partially address this concern, we have rewritten the Results section where we now describe the data of others who have studied the role of TC10 in the secretory pathway. In two papers they found that TC10DN and TC10CA reduced secretion in the range of 40-60% (24), similar to our findings with AMPARs (Figure 1). Using a TC10 FRET sensor construct to assay whether TC10 is in the GTP-TC10 or GDP-TC10 state, they concluded that the TC10 GTPase hydrolysis cycle is required for NPY secretion. TC10DN and TC10CA both reduced secretion by blocking the GTP hydrolysis cycle at different steps. Similar results were obtained using a different secretion assay in PC12 cells (18). Both papers suggested that during exocytosis TC10CA allowed cargo to load into transport vesicles to be delivered to target membranes. TC10CA blocked exocytosis by preventing the GTP-TC10 to GDP-TC10 transition required for transport vesicle docking and/or fusion. In contrast, the results suggest that TC10DN blocked the GDP-TC10 to GTP-TC10 transition, which blocked a different step, cargo loading onto vesicles, thus preventing vesicle delivery to target membranes.

The other way we have addressed this concern was by analyzing in more detail how TC10DN and TC10CA affect the distribution of AMPARs in dendrites (new Figure 2) and comparing the findings to previous studies that have characterized how TC10DN and TC10CA affect the trafficking of other proteins. In new data added to the manuscript, we found that when TC10DN increases the co-localization between GluA1 and Arf6, the GluA1 AMPARs appear to co-localize within a subdomain of the Arf6 fluorescence, consistent with TC10DN blocking AMPAR exit from the Arf6 endosomes. We also found that when TC10CA decreases the co-localization between GluA1 and Arf6 there is an increase in the number of smaller AMPAR puncta (new Figure 2) consistent with an increased number of AMPAR transport vesicles that are not exocytosed.

Also, in addition to monitoring fluorescent exogenous GluA1 it would be worth determining whether the effect of TC10 KO and TC10 mutant-expression holds for endogenous AMPARs as one would expect (using a surface biotinylation protocol, for example).

We performed the experiment requested by the reviewer (new Figure 1). Because our primary interest was the recycling of AMPARs on dendrites, we tested the effects of TC10WT, TC10DN and TC10CA on the endogenous AMPARs on dendrites using an antibody that recognizes an extracellular epitope on GluA1. We obtained results basically identical to the results obtained with the transfected Cherry-GluA1 (compare Figure 1).

The result of the paired recordings is somewhat puzzling. If, as they suggest, TC10 indeed differentiates between surface vs synaptic AMPAR it would be worth considering to determine AMPAR levels in somatic patches (which are expected to be elevated in response to TC10 WT expression).

It is not clear what the reviewer is asking for here. In Figure 1, we have measured the fluorescence of surface AMPARs at the somata and find that TC10WT appeared to have increased GluA1 surface levels but the increase was not statistically significant. It might be possible with somatic outside-out patches to assay to some degree AMPAR levels electrophysiologically with the application of agonist to the patches. In considering it, we felt that this experiment is unlikely to yield significant results because of the difficulties involved with performing it, the possibility that some synaptic receptors may be included in the somatic patches and that we were unable to obtain a significant result using fluorescent methods.

Reviewer #3:

*There are two sorts of data in this paper. One is the effect of a TC10 mutant on the level of GluA1 in a cell compartment or on the extent of a process, such as cLTP. Specifically, the authors express wild type TC10 (WT) and dominant negative (DN) and constitutively active (CA) TC10 mutants and carry out knockdown (KD) of TC10 to dissect GluA1 trafficking. It is confusing that DN and CA expression and KD can have the same effects, decreasing dendritic GluA1 surface levels, while WT has the opposite effect, increasing dendritic GluA1 surface levels (Results section and*
Figure 1). *Why should enhancing and blocking TC10 function have the same result?*

This concern was the same as Reviewer 2's point 1 and addressed above.

The authors state that because the DN and CA mutants reduced surface expression to the same extent, the AMPARs lost from the Arf6 endosomes must be redistributed to an unidentified intracellular compartment. Isn't it possible that overexpression interferes with trafficking non-specifically?

The other is the effect of a mutant on the colocalization of GluA1 with a marker, in particular with transferrin endocytosed by the transferrin receptor. This latter form of data seems more compelling to me and on the basis of such data they authors argue that TC10 functions in GluA1 trafficking associated with Arf6 rather than with the transferrrin receptor, which is a marker for clathrin/dynamin mediate recycling/secretory pathway. The authors conclude that two pathways operate.

Making a case for the operation for the two pathways is important. And for the most part, the data seem respectable. However, the logic for interpretations of the data often escape me, as is the case with similar effects of DN, CA expression and KD of TC10. Also, the authors completely neglect a vast amount of literature that states that GluA1 trafficking is controlled by phosphorylation of the C-terminal domain of the subunit. The authors suggest that TC10 functions in controlling the interaction of nPIST with TARPS that are in complexes with AMPA receptors. But there is abundant evidence that GluA1 phosphorylation, such as takes place during LTP, regulates GluA1 trafficking to and from the surface. The literature does not distinguish which pathway is affected by this phosphorylation (clathrin dependent versus independent), and in this sense the paper contributes a new view. Yet it does so while overlooking the fact that phosphorylation controls the trafficking, be it TC10-dependent or independent.

We completely agree with the reviewer that GluA1 phosphorylation of the C-terminal domain plays an important regulatory role in its recycling and it is very likely to regulate the different events described in this manuscript. We could begin to address this question by altering different kinases and phosphatases, mutating different potential phosphorylation sites on GluA1 or using antibodies specific for the phosphorylated GluA1 protein. Similarly we could address the role of other AMPAR subunit posttranslational modifications such palmitoylation and ubiquitination that have been found to regulate AMPAR recycling. These experiments would definitely begin to characterize in more detail the regulatory mechanisms of the two recycling pathways we describe here and we are very interested in performing them in the future. However, it is not clear to us how these experiments or a discussion of the relevant literature would help answer the basic questions addressed in this manuscript.

Reviewer #4:

Figure 1. *Pictures of neurons only show soma (*Figure 1*) and it would be useful to show the entire neuron, at least in the first figure. Notably, this is all overexpression data and the reader would be enlightened if they also examined endogenous AMPA-R levels. (They do look at endogenous GluA1 later on in terms of localization with known recycling markers). Another concern: discrepancy between electrophysiology and surface labeling of AMPA-Rs for TC10WT vs mutants.*

We agree completely and have added new images that show most of the whole neurons for each of the somas that were displayed in Figure 1 as requested. However, much of the detail described in the Results section is lost because of the reduced sizes of the somata. For this reason and to keep Figure 1 a reasonable size, we have added these images as Figure 1—figure supplement 4.

Figure 2. *Based on title of this figure, the actual supporting data are limited. For instance denditric shaft primary data are only shown for TC10DN and not other conditions, but phenomenon of dendritic accumulation is generalized to all. This is also the only positive data in the figure (other panels are negative) so the authors should be more thorough. The GM130 colocalization data take up most of figure, and doesn't add a lot. Overall it isn't a very thorough investigation of potential effects on forward trafficking.*

Again, we agree completely and removed all of the data from the figures and now display them as Figure 1—figure supplement 4.

Figure 3
*and*
Figure 4. *One concern: DN and CA have different effects on colocalization yet have same effects on AMPA-R accumulation in dendrites?*

This concern was the same as Reviewer 2's point 1 and addressed above.

*Also, the biochemistry in*
Figure 4
*is poorly controlled. The authors should show total protein amounts and also negative control (intracellular protein).*

Both the total protein amounts (inputs) and a negative control using the intracellular protein actin have been added to the figure (new Figure 3) as requested.

Figure 5. *The TC 10DN mutant seems to affect LTD, but not LTP- however the authors follow up on LTP showing that it involves dynamin dependent pathways, while neglecting to follow up on the LTD results. The LTD effect I mostly ignored when summarizing their results overall.*

The reason that we did not perform the same antibody-loading experiments on cLTD as were performed on cLTP is that many previous studies have already found that AMPAR endocytosis is increased during LTP and that it involves the dynamin-dependent pathway. However, the experiments had not been performed on cLTP.

Also, it is essential for the authors to include representative images along with any quantifications.

We agree with the reviewer and have added the representative images to the manuscript as requested. They were not included previously because the images are very similar to those in new Figure 3. Also, old Figure 5 (new Figure 4) is already very large and complicated. For these two reasons we have included the representative images as supplements for Figure 6.

Figure 6. It is nice to have a model figure, but this one is too far reaching. A simpler model included in another figure might be more appropriate.

We agree with the reviewer that a simpler model would be better and have tried to come up with one. To explain all of the data presented in the manuscript the model needs to include: 1) the two recycling pathways, 2) how TC10 WT expression could increase AMPAR surface expression while decreasing AMPAR synaptic levels, 3) a lack of effect of TC10 mutants on LTP and LTD, and 4) how AMPAR endocytosis could increase during both LTP and LTD. We felt that the old model in panel A was needed to contrast with the new model in panel B. We also felt that panels C and D were needed to explain the changes that occur with LTP (panel C) and LTD (panel D). At this point, new Figure 5 is the simplest model that we could come up with that explains all of our results and those of others such as [49].